# Sycp2 is essential for synaptonemal complex assembly, early meiotic recombination and homologous pairing in zebrafish spermatocytes

**Kazumasa Takemoto[1]☯¤, Yukiko Imai [2]☯, Kenji Saito[2], Toshihiro Kawasaki[1,2], Peter M. Carlton [3], Kei-ichiro Ishiguro [4], Noriyoshi Sakai [1,2]***

**1** Department of Genetics, School of Life Science, SOKENDAI (The Graduate University for Advanced Studies), Mishima, Japan, **2** Department of Gene Function and Phenomics, National Institute of Genetics, Mishima, Japan, **3** Radiation Biology Center and Graduate School of Biostudies, Kyoto University, Kyoto, Japan, **4** Department of Chromosome Biology, Institute of Molecular Embryology and Genetics, Kumamoto University, Kumamoto, Japan

☯ These authors contributed equally to this work.
¤ Current address: Laboratory of Chromosome Biology, Institute of Molecular Embryology and Genetics, Kumamoto University, Kumamoto, Japan
* nosakai@nig.ac.jp

**Data Availability Statement:** All relevant data are within the manuscript and its Supporting Information files.

## Abstract

Meiotic recombination is essential for faithful segregation of homologous chromosomes during gametogenesis. The progression of recombination is associated with dynamic changes in meiotic chromatin structures. However, whether Sycp2, a key structural component of meiotic chromatin, is required for the initiation of meiotic recombination is still unclear in vertebrates. Here, we describe that Sycp2 is required for assembly of the synaptonemal complex and early meiotic events in zebrafish spermatocytes. Our genetic screening by N-ethyl-N-nitrosourea mutagenesis revealed that *ietsugu* (*its*), a mutant zebrafish line with an aberrant splice site in the *sycp2* gene, showed a defect during meiotic prophase I. The *its* mutation appeared to be a hypomorphic mutation compared to *sycp2* knockout mutations generated by TALEN mutagenesis. Taking advantage of these *sycp2* hypomorphic and knockout mutant lines, we demonstrated that Sycp2 is required for the assembly of the synaptonemal complex that is initiated in the vicinity of telomeres in wild-type zebrafish spermatocytes. Accordingly, homologous pairing, the foci of the meiotic recombinases Dmc1/Rad51 and RPA, and γH2AX signals were largely diminished in *sycp2* knockout spermatocytes. Taken together, our data indicate that Sycp2 plays a critical role in not only the assembly of the synaptonemal complex, but also early meiotic recombination and homologous pairing, in vertebrates.

## Author summary

Meiosis is an essential type of cell division whose purpose is to produce gametes, such as eggs and sperm, for sexually reproducing eukaryotes. Most cells in the human body are

**Funding:** This work was supported by JSPS KAKENHI (Grant Numbers JP19K16045 and JP18H06057 to YI), and partly supported by JSPS KAKENHI (Grant Number 16H01257 to KI, and 25251034 and 25114003 to NS). The funders had no role in study design, data collection and analysis, decision to publish, or preparation of the manuscript.

**Competing interests:** The authors have declared that no competing interests exist.

diploid cells containing two homologous copies of each chromosome. Meiosis halves the genetic contents of diploid cells and produces haploid gametes with a single copy of each homologous chromosome pair. The faithful transmission of chromosomes is one of the main tasks in meiosis, since extra or missing chromosomes are common causes of genetic disorders, birth defects and infertility. Prior to proper segregation, homologs must pair up and be physically connected. This is achieved by homologous recombination initiated by programmed DNA double-strand breaks. Since it can be deleterious to cells, this initiation step is highly regulated by factors that are not fully understood. In this study, we found that Sycp2, a structural component of meiotic chromatin in metazoans, is essential for the initiation of homologous recombination, meiotic chromatin organization, and homologous pairing in zebrafish. These findings support recent biochemical studies that Sycp2 and Sycp3 serve as a vertebrate counterpart of *S. cerevisiae* Red1.

## Introduction

Meiotic recombination plays a key role in the faithful segregation of homologous chromosomes in the gametogenesis of most sexually reproducing organisms. The key challenge of meiotic recombination is coordination between the processing of recombinant molecules and changes in chromosomal organization to ensure homologous pairing and the formation of at least one crossover per homologous pair. This process is initiated with programmed DNA double-strand breaks (DSBs) catalyzed by the meiotic endonuclease SPO11. Meiotic DSB formation is tightly regulated by several factors, such as the spatiotemporal regulation of chromosome structure and of proteins essential for DSBs.

DSB formation occurs in the context of the chromatin "loop-axis" structure, in which sister chromatids are organized into a series of loops anchored along a proteinaceous axis called the axial element. The loop-axis structure is formed by the localization of axis proteins at early stages of meiotic prophase I, and the axial elements of each homologous pair are connected (synapsed) by a ladder-like structure called the synaptonemal complex (SC) during the progression of meiosis [1]. Observations in budding yeast and in *Caenorhabditis elegans* suggest that axial elements are required for DSB formation. In budding yeast, the axis proteins Hop1 and Red1 are required for efficient DSB formation, since DSB levels are drastically reduced in mutants of these wild-type proteins [2–6]. In *C. elegans*, RNAi-mediated knockdown of the axis component HTP-3 attenuates the formation of foci of the recombinase RAD-51 to the same level as that in *spo-11* null mutants, suggesting that no detectable DSBs are formed in the absence of HTP-3 [7].

One of the primary functions of axis proteins is to recruit proteins essential for DSBs, since Hop1 and Red1 promote axis localization of the Rec114-Mer2-Mei4 complex, which is essential for DSB formation [8]. In vertebrates, many proteins involved in DSB formation and subsequent repair are also observed on the chromosomal axis. For example, mouse REC114, MEI4 and IHO1, orthologs of budding yeast Rec114, Mei4 and Mer2, respectively, localize on chromosomal axes at the leptonema of meiotic prophase I when meiotic DSBs are formed [9–11]. DSB repair proteins, e.g., the single-stranded DNA binding recombinases DMC1 and RAD51, are also observed on chromosomal axes in an SPO11-dependent manner [12–16]. Therefore, the chromosomal axis has been thought to serve as an evolutionarily conserved scaffold for proteins required for meiotic DSB formation.

In metazoans, SYCP2 and SYCP3 compose axal elements that are synapsed by the transverse filament protein SYCP1 [17,18]. A recent biochemical study on axis proteins revealed

that the SC components SYCP2 and SYCP3 form antiparallel heterotetramers that resemble the coiled-coil tetramer structure of budding yeast Red1 [19]. Thus, the SYCP2-SYCP3 complex seems to play a conserved role in meiosis, as a vertebrate counterpart of Red1. However, SYCP3 is dispensable for meiotic DSB formation, as RAD51 foci are still observed in *Sycp3* mutant spermatocytes [20]. In addition, whether Sycp2 is required for meiotic DSB formation has not been elucidated in vertebrates.

SYCP2 was first identified in rats as a component of lateral elements (axial elements after synapsis) of the SC and shows some similarity to the yeast Red1 protein [21]. In rodents, SYCP2 directly interacts with SYCP1 and SYCP3 through its C-terminal domain and internal coiled-coil domain, respectively [22,23]. The deletion of the SYCP3-interacting domain of SYCP2 leads to severe defects in SC formation in mice; SYCP1 is detected as many short fibers, and SYCP3 forms aberrant aggregates [22]. Therefore, SYCP2 is an essential component for proper SC formation in mice.

Sycp2 and key meiotic features, such as the SC structure and crossover by recombination, are also conserved in zebrafish meiosis [24–26]. Previously, we isolated male sterile zebrafish lines generated by N-ethyl-N-nitrosourea (ENU) mutagenesis [25]. One of these mutant lines, *ietsugu* (*its*), shows defects in the assembly of Sycp3 during prophase I [25,27]. In this study, we mapped the *its* mutation to the zebrafish *sycp2* gene and showed that it is a hypomorphic mutation compared to the *sycp2* knockout mutation. Our data indicated that zebrafish Sycp2 has a conserved function in SC formation at early prophase I. We further demonstrated that DSB signals and homologous pairing were strongly diminished in *sycp2* knockout spermatocytes. Altogether, these results indicate that Sycp2 has critical functions in early meiotic recombination and homologous pairing as well as the SC assembly, as reported for budding yeast Red1.

## Results

### The *sycp2* gene is responsible for *its* mutant phenotypes

To map a genomic locus responsible for the sterility of the *its* mutant line, we performed screening with simple sequence-length polymorphism (SSLP) markers (see Materials and Methods). The causal genomic region was mapped on a 5.5 Mbp region on chromosome 23 containing 77 annotated open reading frames, including that of the *sycp2* gene (S1 Table). We have previously reported that *its* spermatocytes show aggregation of the Sycp3 protein in the nucleus (see below) [25]. Because this phenotype resembles that of *Sycp2* mutant spermatocytes in mice [22], we first examined the coding sequence of the *sycp2* gene; however, no mutation was detected. We next examined the expression of *sycp2* mRNA in *its* mutant and wild-type testes by RT-qPCR (Fig 1A). The expression of *sycp2* in *its* mutant testes was only 25.9% of that in wild-type testes (*p<0.01*). Thus, the expression of *sycp2* mRNA was significantly reduced in *its* mutant testes.

To examine whether the decreased expression of *sycp2* is responsible for the *its* phenotypes, we generated *sycp2* knockout zebrafish by TALEN mutagenesis targeting exon 18 (S1 Fig). We isolated three deletion mutations: two frameshift mutations (14- and 16-bp deletions; $sycp2^{\Delta14}$ and $sycp2^{\Delta16}$, respectively) and a mutation without a frameshift (3-bp deletion; $sycp2^{\Delta3}$). As the phenotypes of the two frameshift mutant fish were identical at cytological levels (S6 Fig, S7 Fig), the $sycp2^{\Delta14}$ allele was used as the *sycp2* knockout allele (*sycp2⁻*) in this study, unless otherwise noted. Notably, homozygote mutant females were obtained for the $sycp2^{\Delta3}$ allele, but not for the $sycp2^{\Delta14}$ and $sycp2^{\Delta16}$ alleles, after intercrossing of $F_1$ heterozygous fish. This is consistent with our previous observation that no females appear among *its* mutant fish (see below) [25]. Therefore, subsequent analyses of the *sycp2* mutant were performed only with male zebrafish. We confirmed the depletion of *sycp2* mRNA expression in $sycp2^{-/-}$ testes by RT-qPCR

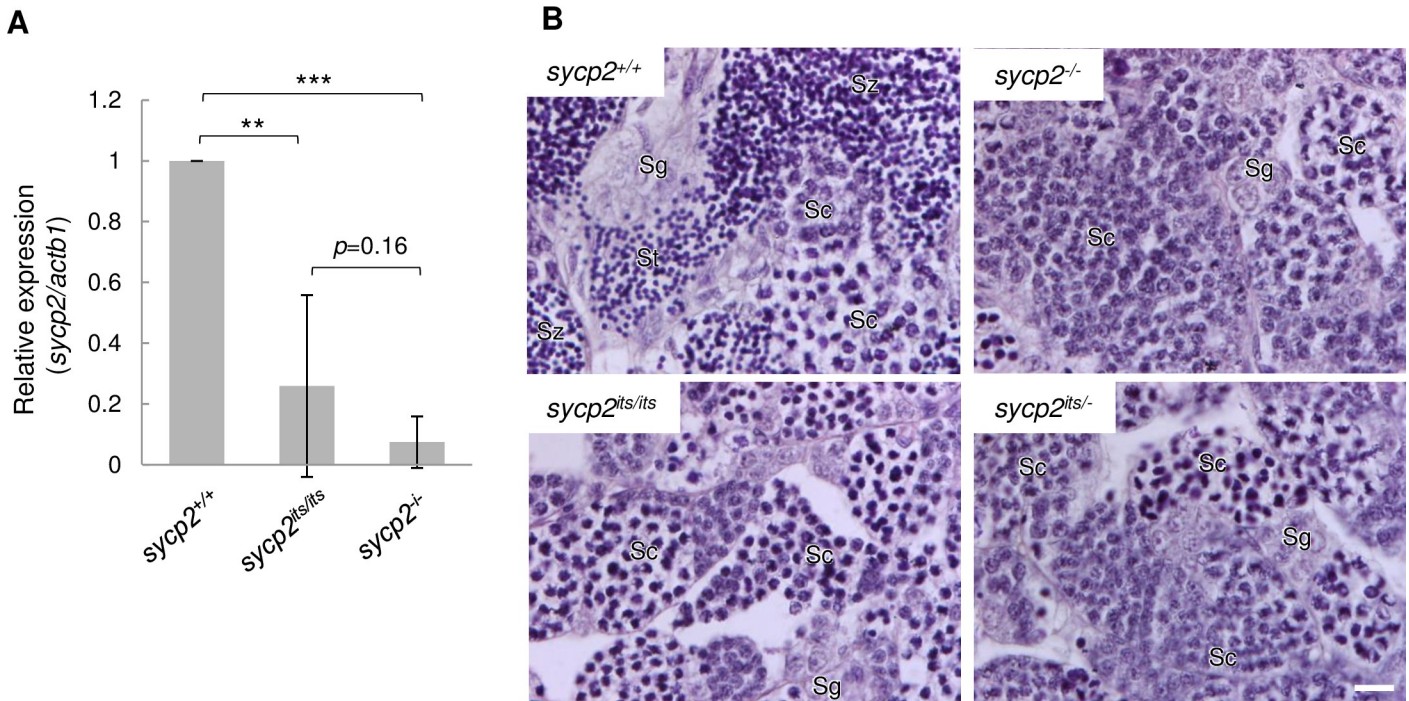

**Fig 1. *its* mutant phenotypes are associated with the *sycp2* gene.** A: Expression of *sycp2* mRNA in *sycp2*<sup>+/+</sup>, *sycp2*<sup>its/its</sup> and *sycp2*<sup>-/-</sup> testes. The mRNA levels relative to *actb1* are shown (n = 5). The vertical bars indicate the SD. ** and *** indicate $p<0.01$ and $p<0.001$, respectively (Student's t-test). B: HE-stained sections of *sycp2*<sup>+/+</sup>, *sycp2*<sup>-/-</sup>, *sycp2*<sup>its/its</sup>, and *sycp2*<sup>its/-</sup> testes. All samples were prepared from the same siblings. Sg: spermatogonia, Sc: spermatocytes, St: spermatids, Sz: spermatozoa. Spermatids and spermatozoa were observed only on the *sycp2*<sup>+/+</sup> section. Scale bar, 10 µm.

(Fig 1A) and of the Sycp2 protein by Western blotting (S5 Fig), showing that the frameshift mutation resulted in a null phenotype for the *sycp2*<sup>Δ14</sup> allele presumably through nonsense-mediated mRNA decay, a eukaryotic pathway to eliminate transcripts with a premature stop codon (see below) [28]. We examined the fertility of *sycp2*<sup>-/-</sup> male fish after allowing natural mating with wild-type female fish. No fertilized eggs were identified after examining 825 eggs from six crosses between *sycp2*<sup>-/-</sup> male and wild-type female fish, while 66.4% of eggs (249 of 375 eggs) from four crosses between wild-type male and female fish developed normally into gastrulae. Consistent with this result, histological analysis demonstrated that neither spermatids nor spermatozoa were present in *sycp2*<sup>-/-</sup> testes (Fig 1B; *sycp2*<sup>-/-</sup>). We obtained fertilized eggs from the mating of *sycp2*<sup>Δ3/Δ3</sup> males and wild-type females (82.1%, 348 of 424 eggs from four crosses). Therefore, it can be ruled out that the sterility and meiotic phenotypes (see below) observed in *sycp2*<sup>-/-</sup> fish are caused by off-target effects on another locus.

Next, we examined whether the *its* mutation can complement the sterility caused by the *sycp2*<sup>-</sup> mutation. After mating of *its* heterozygous mutant fish (*sycp2*<sup>its/IM</sup>; IM is an inbred wild-type strain derived from *India*; see Materials and Methods) and *sycp2*<sup>+/-</sup> fish, we obtained 67 offspring, among which 17 fish were *sycp2*<sup>its/-</sup>. Histological analysis of the *sycp2*<sup>its/-</sup> testes showed similar phenotypes as those observed in *sycp2*<sup>-/-</sup> and *sycp2*<sup>its/its</sup> testes, where neither spermatids nor spermatozoa, but accumulations of spermatocytes were observed (Fig 1B). After mating five *sycp2*<sup>its/-</sup> males with wild-type females, no fertilized eggs were observed among 1785 eggs examined, while 31.4% of embryos from a *sycp2*<sup>IM/-</sup> sibling cross (123 of 391 eggs) and 94.9% of embryos from a wild-type sibling cross (227 of 239 eggs) were fertilized and developed normally. Thus, the *sycp2* knockout allele could not complement the *its* phenotypes. This result suggests that meiotic defects in *its* testes are caused by reduced expression of Sycp2.

## Aberrant splicing of *sycp2* mRNA is caused by the *its* mutation

To identify the potential mutation responsible for low *sycp2* expression, we examined the sequence of *sycp2* cDNA prepared from *its* and wild-type testes. After sequencing of several *sycp2* cDNAs from *its* individuals, we found an insertion of 5 bp at the splice site between exons 8 and 9 in addition to the wild-type sequence (Fig 2A and S2 Fig). This 5-bp insertion was not detected in *sycp2* cDNA prepared from wild-type testes. This finding implies that an abnormal splicing event occurs in *its* mutants. Therefore, we examined the genomic sequences of *sycp2* intron 8 and identified a T-to-A substitution of the 127th residue in the *its* but not in the wild-type genome (Fig 2B). This substitution generates a potential aberrant splice site upstream of the canonical splice site of exons 8 and 9. We performed RT-PCR with a primer pair flanking the exon 8–9 junction of *sycp2* to determine how much of the *sycp2* mRNA was spliced at the abnormal site. Strikingly, most of the *its* mRNA was spliced at the aberrant site in intron 8 (Fig 2C), and only ~15% of *sycp2* mRNA was found to be spliced at the canonical site in *its* at exons 8–9 after sequencing of *sycp2* cDNA (S2 Table).

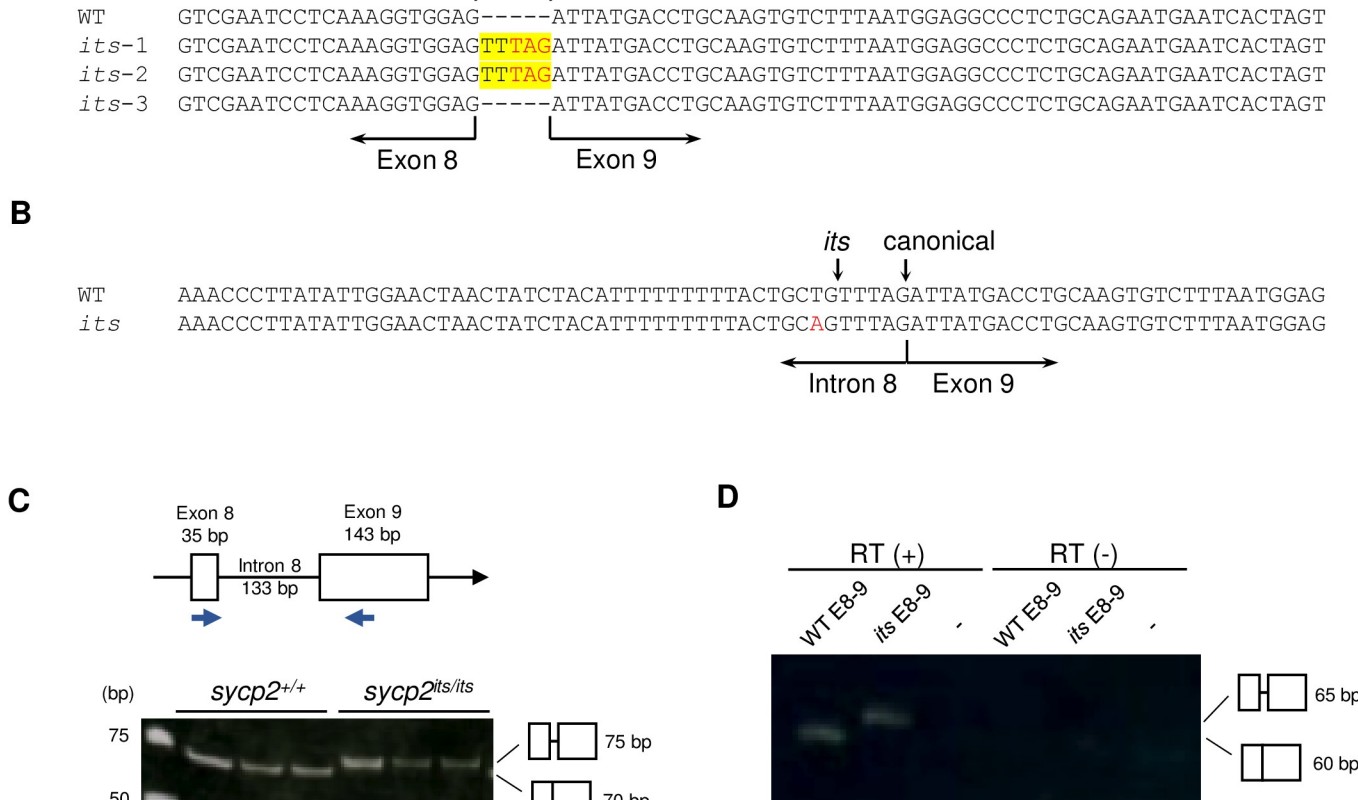

**Fig 2. Aberrant splicing of *sycp2* mRNA in *sycp2^its/its* testes.** A: cDNA sequences of the exon 8–9 junction of *sycp2*. The sequences of one *sycp2^+/+* (WT) and three *sycp2^its/its* (*its*-1 to *its*-3) cDNA clones are shown. Two of the three *sycp2^its/its* cDNA clones had a 5-bp insertion (in yellow) containing a premature stop codon (in red letters) at the exon 8–9 junction. B: Genomic sequences of the intron 8-exon 9 junction of *sycp2*. Sequences from *sycp2^+/+* (WT) and *sycp2^its/its* (*its*) testes are shown. The *sycp2^its* allele harbors a T-to-A substitution (in red) upstream of exon 9. The *its* and wild-type (canonical) splicing sites are indicated with vertical arrows. C: Splicing at *sycp2* exon 8-exon 9 was examined by RT-PCR. The target sites of the PCR primers are shown above. Template cDNA was prepared from *sycp2^+/+* (WT) and *sycp2^its/its* (*its*) testes from three individual fish. The PCR products were examined on a 20% acrylamide gel. D: Mini-gene splicing assay in the wild-type genetic background. RT-PCR was performed with caudal fin cDNA from transgenic fish with either wild-type (WT E8-9) or *its*-type (*its* E8-9) *sycp2* mini-genes (S3 Fig). Wild-type fish without mini-genes were used as controls (-). See S3 Fig for more clones and the results of control PCR with a primer pair specific to EGFP.

Finally, to verify that this abnormal splicing was caused by the T-to-A substitution and not by a trans-factor in the *its* line, we carried out a mini-gene splicing assay by introducing mini-gene constructs into wild-type zebrafish (S3 Fig and Fig 2D, see Materials and Methods). RT-PCR of the splice site sequence transcribed from the mini-gene construct revealed that the T-to-A substitution in *its sycp2* intron 8 was sufficient to introduce aberrant splicing in the wild-type context (Fig 2D). Notably, the 5-bp sequence inserted in the *sycp2* cDNA of *its* contains a premature terminal TAG codon (Fig 2A). In eukaryotes, mRNAs that contain premature stop codons are often eliminated by nonsense-mediated mRNA decay [28]. Thus, the low level of *sycp2* expression in *its* testes potentially results from the insertion of a premature termination codon by aberrant exon 8–9 splicing, corroborating the above assumption. Altogether, these results suggest that *its* phenotypes are primarily derived from mis-splicing resulting from a T-to-A substitution in the *sycp2* intron 8.

## No female appears among *sycp2*$^{-/-}$ zebrafish

As mentioned above, no females were observed among *sycp2*$^{-/-}$ knockout fish, as previously reported for *its* mutant fish [25]. This phenotype was markedly different from *spo11* mutant zebrafish, which are deficient in meiotic DSB formation; *spo11*$^{-/-}$ females generate eggs capable of fertilization, but the resulting embryos develop abnormally, most likely due to aneuploidy [26]. To confirm the sexual phenotypes of *sycp2* knockout zebrafish and to compare them with those of *spo11* knockout zebrafish, we analyzed genotypes of male and female offspring from the intercrossing of each line (Table 1). To determine the sexual phenotypes, we examined the morphology of gonads dissected from 8- to 9-week-old offspring (see Materials and Methods). We observed homozygote mutant males in offspring from both *sycp2*$^{+/-}$ and *spo11*$^{+/-}$ intercrossing (31% and 17% of all male offspring, respectively). While 24% (11 of 46 fish) of female offspring from the *spo11*$^{+/-}$ intercrossing appeared to be homozygous mutant, no homozygous mutant female appeared among offspring from the *sycp2*$^{+/-}$ intercrossing. In wild-type zebrafish, all juveniles develop gonads with immature oocytes regardless of their definitive sex, and the individuals in which these immature oocytes degenerate become males [29–31]. Therefore, zebrafish mutants depleted of oocytes develop as males that are mostly infertile [32–35]. In *sycp2*$^{-/-}$ juvenile gonads, we observed previtellogenic dictyate oocytes (late stage IB oocytes [36]) in four among five individual fish at 28 days postfertilization (S4 Fig). Therefore, the absence of females in *sycp2*$^{-/-}$ fish (Table 1) implies that oocytes were eliminated at the juvenile age or later, possibly due to checkpoint activation in oogenesis. As a result, all *sycp2* mutant fish develop as males.

## SC formation is impaired in *sycp2* mutant spermatocytes

SYCP2 is a conserved lateral element component of the SC in metazoans and is essential for SC formation in mice [18,22]. We previously reported abnormal aggregation of Sycp3 in *its* spermatocytes [25], suggesting that zebrafish Sycp2 has a conserved function in axis and SC formation. To further understand Sycp2 functions in SC assembly, wild-type and *sycp2* mutant

**Table 1. Quantification of sexual phenotypes of *spo11* and *sycp2* mutant zebrafish.**

| Mating | Gonad morphology | +/+ | +/- | -/- | Total |
|---|---|---|---|---|---|
| *sycp2+/- x sycp2+/-* | Testis | 14 (16%) | 47 (53%) | 27 (31%) | 88 |
| | Ovary | 25 (54%) | 21 (46%) | 0 (0.0%) | 46 |
| *spo11+/- x spo11+/-* | Testis | 31 (33%) | 47 (50%) | 16 (17%) | 94 |
| | Ovary | 14 (30%) | 21 (46%) | 11 (24%) | 46 |

spermatocyte chromosomal spreads were immunostained with antibodies against the SC components Sycp3, Sycp2 and Sycp1 (Fig 3 and S6 Fig). In wild-type spermatocytes, Sycp3 appeared on chromatin prior to Sycp2 and Sycp1, as a few bright foci at preleptonema (Fig 3A-i). Sycp3 began to be extended as short fragments upon the appearance of Sycp2 signals at leptonema (Fig 3A-ii). At zygonema, Sycp1 began to be observed as short fragments, following the appearance of Sycp3 and Sycp2 signals (Fig 3A-iii). All the Sycp3, Sycp2, and Sycp1 fragments extended during the progression of zygonema (Fig 3A-iv), and 25 pairs of homologous chromosomes were stained for these three proteins along the entire length at pachynema (Fig 3A-v).

In *sycp2*<sup>-/-</sup> spermatocytes, Sycp3 was detected only as a few bright aggregates similar to those in wild-type preleptotene nuclei (Fig 3B-i), suggesting that Sycp2 is required for the localization and extension of Sycp3 along the axis in zebrafish spermatocytes. Furthermore, despite the absence of Sycp2 and Sycp3 fragments, *sycp2*<sup>-/-</sup> spermatocytes showed aberrant filaments of Sycp1 that were not homogeneous in length, and their numbers were drastically decreased compared to those in wild-type nuclei (Fig 3B-i and S6 Fig). Thus, *sycp2*<sup>-/-</sup> spermatocytes are arrested at zygotene-like stages, with aberrant axis morphology and SC. In the majority of *sycp2*<sup>its/its</sup> spermatocytes, Sycp3 was observed as aggregates, and Sycp2 was hardly detected, as observed in *sycp2*<sup>-/-</sup> nuclei (Fig 3B-ii to 3B-iv). However, we noticed that a subpopulation of *sycp2*<sup>its/its</sup> spermatocytes showed Sycp1 staining patterns that were intermediate between those of wild-type and *sycp2*<sup>-/-</sup> nuclei, showing more Sycp1 filaments than *sycp2*<sup>-/-</sup> nuclei (Fig 3B-i to 3B-iv, S6 Fig). In a limited population of *sycp2*<sup>its/its</sup> spermatocytes, we also observed nuclei that showed pachytene-like appearances with regard to the staining patterns of Sycp3, Sycp2 and Sycp1 (Fig 3B-v). Since such nuclei were found in only one male out of 4 individual *sycp2*<sup>its/its</sup> males at a rate of 3.8% of Sycp1-positive nuclei (5 among 133 nuclei), this heterogeneity of the *sycp2*<sup>its/its</sup> phenotype among cells and individuals might have been derived from the stochastic nature of a splicing event caused by the *its* mutation. Thus, the intermediate phenotypes of *sycp2*<sup>its/its</sup> between the wild-type and *sycp2*<sup>-/-</sup> phenotypes indicate that *sycp2*<sup>its</sup> is a hypomorphic allele. Taken together, these results demonstrate that Sycp2 is essential for SC formation in zebrafish.

## Ectopic localization of Sycp1 in *sycp2* mutant spermatocytes

In wild-type zebrafish spermatocytes, the SC emanates from telomeres (S7 Fig) [26,27]. As Sycp1 filaments were abnormal in number and length in *sycp2* mutant zebrafish spermatocytes (Fig 3B and S6 Fig), we next sought to determine whether these aberrant Sycp1 filaments were associated with telomeres using a telomere-targeting polyamide (see Materials and Methods) [37]. For this purpose, the proportions of Sycp1 filaments colocalizing with telomere foci at chromosomal ends were quantitated in wild-type and *sycp2* mutant spermatocytes (Fig 4A and 4B). In wild-type spermatocytes, most Sycp1 filaments colocalized with telomere foci at either end in zygonema (Fig 4A-i and 4B: 96%±4.9 (SD)) and at both ends in pachynema (Fig 4A-ii and 4B: 97%±4.1). In contrast, in *sycp2*<sup>-/-</sup> spermatocytes, only ~25% of Sycp1 filaments colocalized with telomeres at chromosomal ends (Fig 4A-iv and 4B). These results indicate that in the absence of Sycp2, the majority of Sycp1 filaments extend at interstitial sites apart from telomeres. Notably, these aberrant Sycp1 filaments were costained with the meiotic cohesin Rad21l1, suggesting that Sycp1 could be recruited onto chromatin in the absence of Sycp2 (S8 Fig). In *sycp2*<sup>its/its</sup> spermatocytes, the number of Sycp1 filaments colocalized with telomeres was reduced (76%±18.5) compared to that in wild-type spermatocytes (Fig 4A-v and 4B), supporting the above assumption that *sycp2*<sup>its</sup> is a hypomorphic allele. Altogether, these data indicate that Sycp2 promotes the localization of Sycp1 at telomeres.

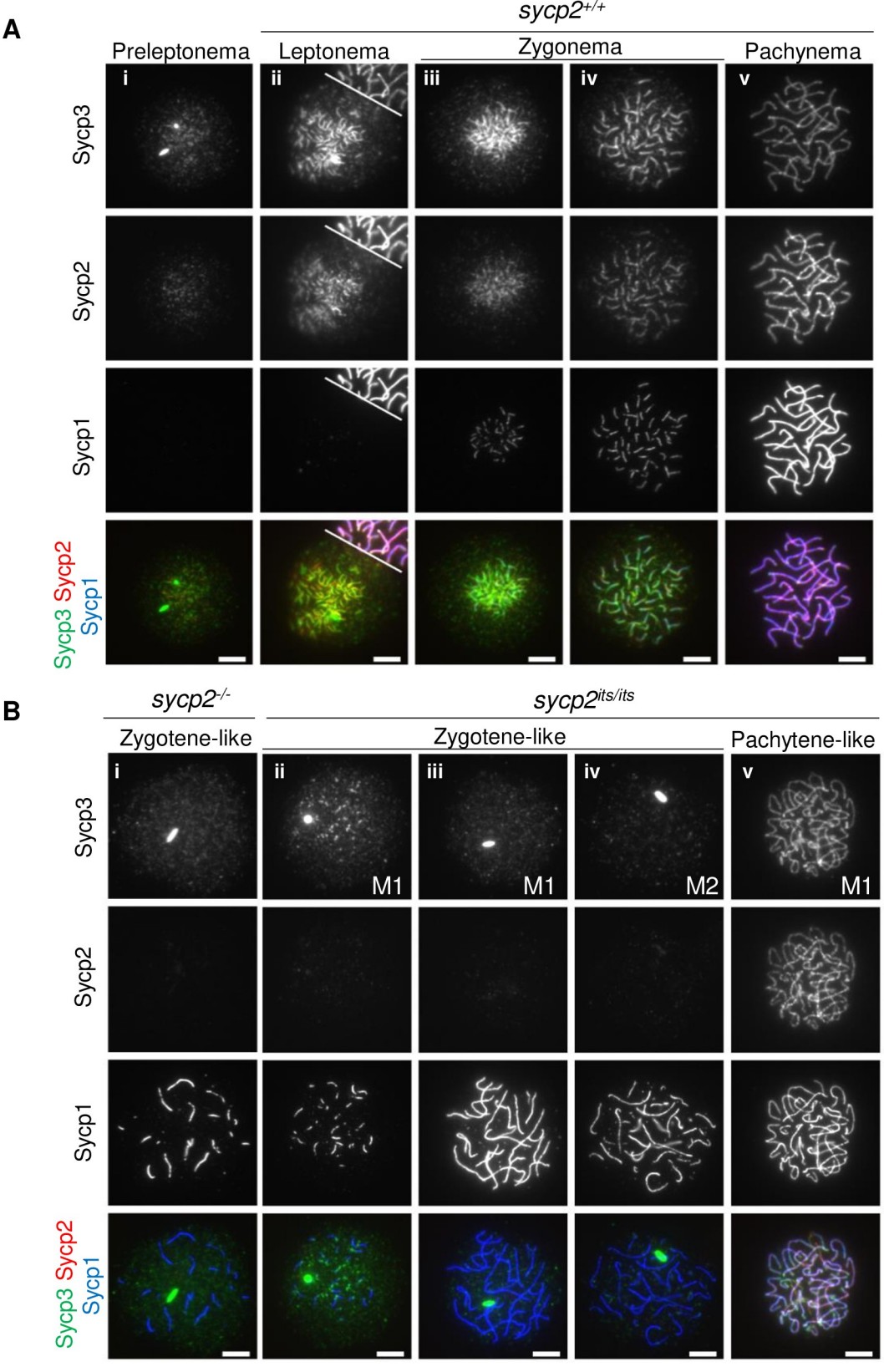

**Fig 3. SC formation is impaired in *sycp2* mutant spermatocytes.** Immunostaining of SC components on wild-type (A), *sycp2*$^{-/-}$ and *sycp2*$^{its/its}$ (B) spermatocyte chromosomal spreads. Individual images with anti-Sycp3, anti-Sycp2, or anti-Sycp1 antibodies and a merged image are shown for each nucleus. Wild-type nuclei are staged according to the staining patterns of Sycp3, Sycp2 and Sycp1 (A-i to A-v) [26]. The white line on A-ii indicates a border with another nucleus on the top right. Nuclei from two individual *sycp2*$^{its/its}$ males (M1 and M2) are shown (B-ii to B-v). Wild-type-like nuclei were observed only among M1 spermatocytes. All spreads were prepared and stained at the same time, and all images were processed in the same manner. Scale bars, 5 μm.

## Homologous pairing is attenuated in *sycp2* mutant spermatocytes

When we examined telomere foci, we noticed that a greater number of telomere foci were present in *sycp2* mutant spermatocytes than in wild-type spermatocytes (Fig 4A). Since telomeres are located at both ends of the 25 paired zebrafish homologous chromosomes, 50 telomere foci per nucleus are expected to be observed after the completion of pairing in normal pachytene spermatocytes, whereas the presence of more than 50 is assumed to indicate a pairing defect in zygotene-like spermatocytes in the absence of Sycp2. To examine this idea, the numbers of telomeres were quantified and compared in wild-type, *sycp2*$^{its/its}$, and *sycp2*$^{-/-}$ spermatocytes, which exhibit Sycp1 filaments (Fig 4C). In wild-type nuclei, the average number of telomere foci per nucleus was 51±4.8, indicating that almost all 25 chromosome pairs were paired at the chromosomal ends. Nuclei with fewer than 50 telomere foci were at early zygonema, in which telomere foci are observed as a cluster, causing an underestimation of the number of foci. In *sycp2*$^{its/its}$ and *sycp2*$^{-/-}$ zygotene-like spermatocytes, the average numbers of telomere foci per nucleus were 69±9.0 and 94±16, respectively, and were significantly higher than those in wild-type nuclei ($p<0.0001$). These results indicate that chromosome pairing is severely impaired in the absence of Sycp2 and that this phenotype is moderate in *sycp2*$^{its/its}$ spermatocytes.

Next, we further examined whether pairing at homologous sites is defective in *sycp2* mutant spermatocytes by using a BAC probe that recognizes an ~68 kbp region on chromosome 5 [26]. After costaining of Sycp1 with the BAC probe, pairing of BAC foci were examined in Sycp1-positive nuclei (Fig 5). When BAC signals were observed as one or two foci with an intervening distance of <3 μm in a nucleus, they were considered to be paired. In wild-type fish, the majority of nuclei were stained with paired BAC foci (88%; Fig 5A-i and 5B), while 12% of nuclei were stained with two unpaired foci (Fig 5A-ii and 5B). Thus, the BAC-stained locus was mostly paired in wild-type zygotene and pachytene spermatocytes. Strikingly, the proportion of Sycp1-positive nuclei with paired BAC foci was reduced to 51% and 8% in *sycp2*$^{its/its}$ and *sycp2*$^{-/-}$ spermatocytes, respectively (Fig 5A-iii, 5A-iv and 5B). Instead, the majority of Sycp1-positive nuclei were stained with unpaired BAC foci separated from each other (an intervening distance of >3 μm) in these genotypes (Fig 5A-v, 5A-vi and 5B). Some of such unpaired foci were observed as three or four foci, in a small fraction of *sycp2* mutant nuclei (Fig 5A-vii and 5B). Therefore, homologous pairing at the BAC-stained locus appeared to be affected in *sycp2* mutant spermatocytes. This phenotype was more pronounced in *sycp2*$^{-/-}$ spermatocytes, in which 92% of nuclei were stained with unpaired BAC foci (Fig 5A-vi, 5A-vii and 5B). Altogether, these results showed that homologous pairing is severely impaired in the absence of Sycp2.

## *sycp2*$^{-/-}$ zebrafish spermatocytes are defective in early meiotic recombination

Sycp2 composes the chromosomal axis, which is thought to be a site for meiotic DSB formation and repair. To examine DSB formation in zebrafish, we generated a guinea pig antiserum for zebrafish Dmc1 (S5 Fig), an evolutionarily conserved meiotic protein that binds to single-

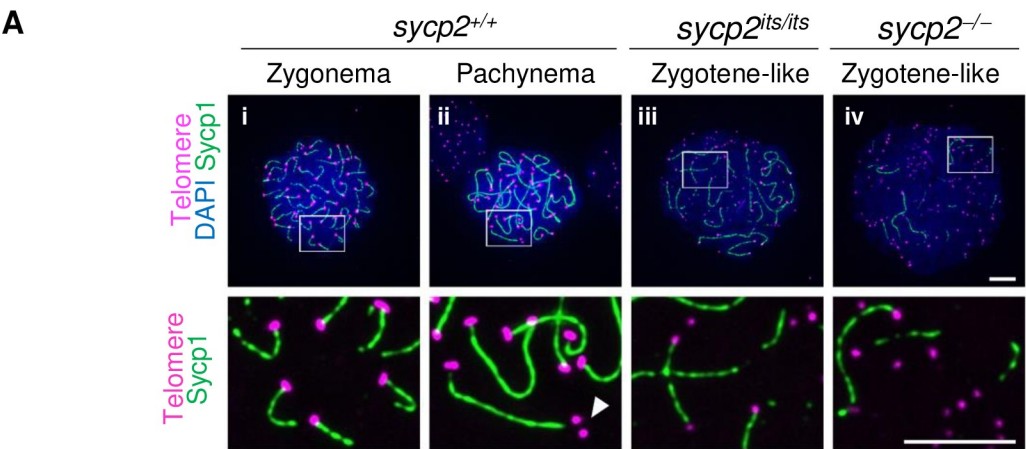

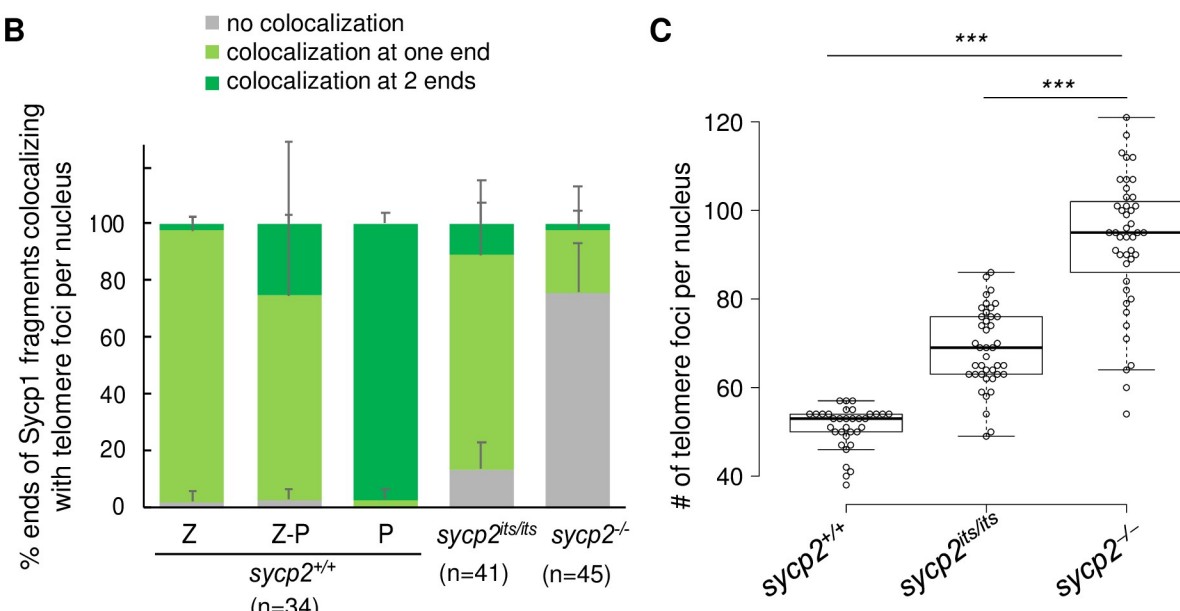

**Fig 4. Ectopic localization of Sycp1 and paring defects at telomeres in *sycp2* mutant spermatocytes.** A: Costaining of telomeres and Sycp1 on *sycp2*⁺/⁺ (i-ii), *sycp2*^its/its (iii), and *sycp2*⁻/⁻ (iv) spermatocyte chromosomal spreads. The regions outlined with white are shown at a higher magnification at the bottom. The nuclei of *sycp2*⁺/⁺ spermatocytes are at zygonema (i; telomere detected at one end of each Sycp1 filament) and pachynema (ii; telomeres detected at both ends of Sycp1 filaments). Scale bars, 5 μm. B: Colocalization of Sycp1 fragments with telomere foci stained by the telomere-targeting polyamide at their ends in *sycp2*⁺/⁺, *sycp2*^its/its and *sycp2*⁻/⁻ spermatocytes. The percentages of Sycp1 filaments that did not colocalize or that colocalized at one or both ends with telomeres are shown in gray, light green and green, respectively. Each pool corresponds to the nuclei counted in S6 Fig. Error bars indicating the SD are shown only for the plus direction. A small fraction of Sycp1 ends did not colocalize with telomere foci, but two telomere foci were observed in close proximity (Fig 4A, arrowhead). These sites were possible locations at which synapsis or desynapsis was in progress. C: Numbers of telomere foci in *sycp2*⁺/⁺, *sycp2*^its/its and *sycp2*⁻/⁻ spermatocytes. Telomere foci stained by the telomere-targeting polyamide were counted in nuclei with Sycp1 filaments of each genotype. Center lines show the medians; box limits indicate the 25th and 75th percentiles as determined by R software; whiskers extend 1.5 times the interquartile range from the 25th and 75th percentiles; data points are plotted as open circles. Chromosomal spreads of one (wild-type) or two (*sycp2*^its/its and *sycp2*⁻/⁻) individual fish were used for counting. *sycp2*⁺/⁺, n = 34; *sycp2*^its/its, n = 41; and *sycp2*⁻/⁻, n = 45. *** indicates *p*<0.0001 (Student's t-test).

stranded DNA generated at DSB sites [38,12,13]. It should be noted that our anti-Dmc1 antiserum potentially recognizes another recombinase, Rad51, since they share high similarity at the amino acid sequence level (see Materials and Methods). In wild-type spermatocytes, many

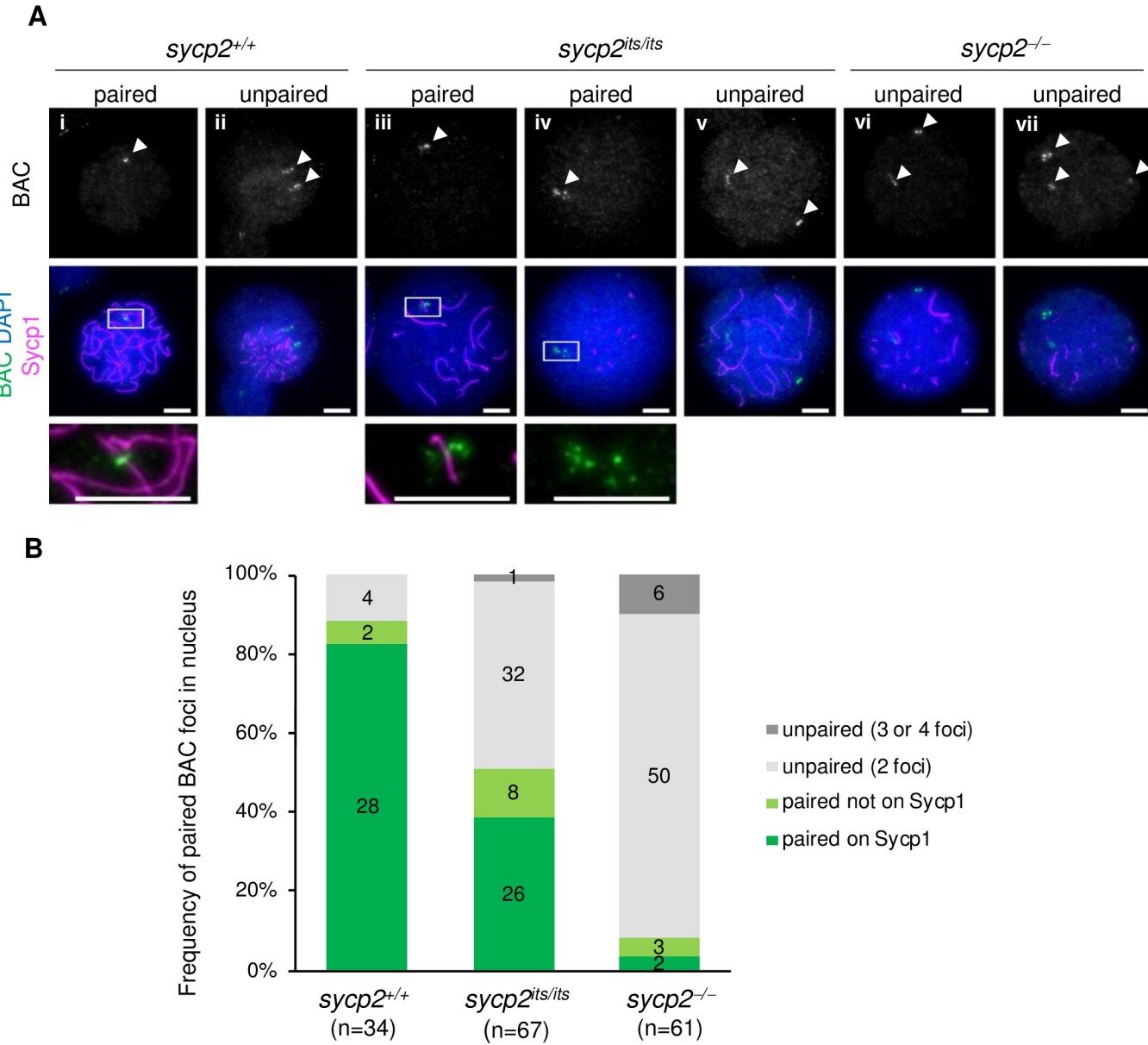

**Fig 5. Homologous pairing is impaired in *sycp2* mutant spermatocytes.** A: Fluorescent in situ hybridization with a BAC probe in *sycp2*<sup>+/+</sup>, *sycp2*<sup>its/its</sup> and
*sycp2*<sup>-/-</sup> spermatocytes. The regions outlined with white in A-i, A-iii and A-iv are shown at a higher magnification at the bottom. White arrowheads indicate
BAC-probe-stained foci. A-i, A-iii and A-iv show nuclei with paired BAC foci with (A-i and A-iii) and without (A-iv) colocalization on an Sycp1 fragment. A-ii,
A-v, A-vi and A-vii show nuclei with multiple unpaired BAC foci. Scale bars, 5 μm. B: Quantification of pairing of BAC foci in *sycp2*<sup>+/+</sup> (n = 34), *sycp2*<sup>its/its</sup>
(n = 67) and *sycp2*<sup>-/-</sup> (n = 61) spermatocytes. The percentage of nuclei stained for one to four BAC foci is shown for each genotype. When BAC signals in a
nucleus were observed as one focus or two foci with an intervening distance of <3 μm, they were considered to be paired, a localization on Sycp1 filaments was
also evaluated. Proportions corresponding to one BAC focus with localization (paired on Sycp1) and without localization (paired not on Sycp1) on Sycp1
filaments are shown in dark and light green, respectively. Chromosomal spreads of one (wild-type) or two (*sycp2*<sup>its/its</sup> and *sycp2*<sup>-/-</sup>) individual fish were used for
counting.

bright Dmc1/Rad51 foci localized on Sycp3-stained axes in close proximity to telomere foci
from leptonema to zygonema (Fig 6A-i to 6A-iii, S9 Fig). To determine whether Dmc1/Rad51
foci were formed in a Spo11-dependent manner in zebrafish spermatocytes, a *spo11* knockout
zebrafish line was generated by CRISPR-Cas9 mutagenesis (S10 Fig). In contrast to the case in
wild-type spermatocytes, Dmc1/Rad51 signal was rarely detected in *spo11*<sup>-/-</sup> spermatocytes at

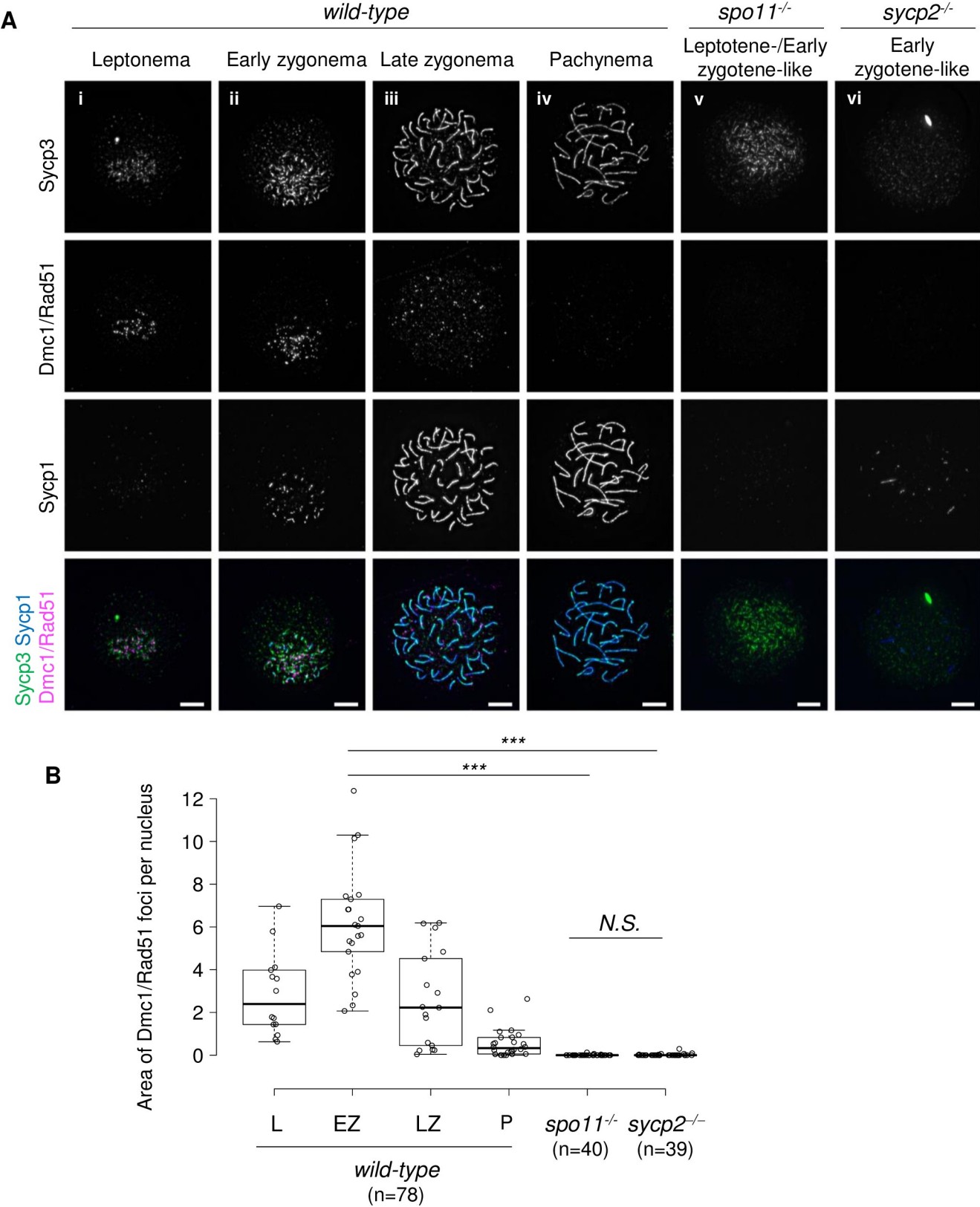

**Fig 6. Formation of Dmc1/Rad51 foci is impaired in *sycp2*^(-/-) spermatocytes.** A: Immunostaining of Dmc1/Rad51, Sycp1 and Sycp3 on wild-type (i to iv), *spo11*^(-/-) (v) and *sycp2*^(-/-) (vi) spermatocyte chromosomal spreads. The wild-type nuclei are at leptonema (i), early zygonema (ii), late zygonema (iii) and

pachynema (iv) according to the Sycp1 and Sycp3 staining patterns. A *spo11*−/− nucleus at an early zygotene-like stage, according to Sycp3 staining patterns, is shown (v). An early zygotene-like *sycp2*−/− nucleus stained with short Sycp1 fragments is shown (vi). Scale bars, 5 μm. B: Quantification of the Dmc1/Rad51-stained area. The sum of the area stained for the Dmc1/Rad51 foci in each nucleus was measured in wild-type nuclei at leptonema (L, n = 14), early zygonema (EZ, n = 21), late zygonema (LZ, n = 17), and pachynema (P, n = 26), in leptotene- or early zygotene-like *spo11*−/− (n = 40), and in early zygotene-like *sycp2*−/− (n = 39) spermatocytes. Center lines show the medians; box limits indicate the 25th and 75th percentiles as determined by R software; whiskers extend 1.5 times the interquartile range from the 25th and 75th percentiles; data points are plotted as open circles. Chromosomal spreads of two individual fish were used for each genotype. *** indicates $p<0.0001$ (Student's *t*-test). *N.S.* indicates not significant.

the leptotene-/early zygotene-like stage (Fig 6A-v). These results are consistent with those of a recent study showing that Rad51, a marker of DSBs, forms foci adjacent to telomeres in a Spo11-dependent manner [26].

Next, we examined meiotic DSB formation in *sycp2*−/− spermatocytes by Dmc1/Rad51 staining (Fig 6-vi). As *sycp2*−/− spermatocytes cannot be staged by Sycp3 patterns, nuclei with short Sycp1 filaments (early zygotene-like) were compared to wild-type nuclei at early zygonema. Remarkably, Dmc1/Rad51 foci were rarely detected in early zygotene-like *sycp2*−/− nuclei (Fig 6A-vi). These nuclei were in striking contrast to wild-type nuclei of early zygonema, in which bright signals of Dmc1/Rad51 were observed (Fig 6A-ii). This result is similar to what was observed in *spo11*−/− spermatocytes (Fig 6A-v). To further compare the loss of Dmc1/Rad51 foci in *sycp2*−/− and *spo11*−/− spermatocytes, we quantified Dmc1/Rad51 signals (Fig 6B). Because many Dmc1/Rad51 foci cluster around the telomere bouquet in zebrafish spermatocytes, counting numbers of discrete Dmc1/Rad51 foci is difficult. Thus, the total area stained with Dmc1/Rad51 foci in each nucleus was measured to estimate the numbers of Dmc1/Rad51 foci per spermatocyte. As a result, both *sycp2*−/− and *spo11*−/− spermatocytes showed a significant decrease in Dmc1/Rad51 signals compared to wild-type spermatocytes at early zygonema ($p<0.0001$, Fig 6B). In contrast, we could not observe a significant difference between Dmc1/Rad51 signals of *sycp2*−/− and *spo11*−/− spermatocytes ($p = 0.3176$). Therefore, the formation of Dmc1/Rad51 foci was impaired in the absence of Sycp2 at a similar level to that in *spo11*−/− spermatocytes.

These results suggest that meiotic DSBs are not formed or that these recombinase proteins are unable to localize on DSB sites in the absence of Sycp2. To examine these two possibilities, we examined formation of Dmc1/Rad51 foci in *sycp2*−/− spermatocytes, after exogenous induction of DSBs by γ-ray irradiation (S11 Fig). Spermatocyte chromosomal spreads were prepared after 10 Gy γ-ray irradiation of wild-type, *spo11*−/− and *sycp2*−/− zebrafish, and were stained for Dmc1/Rad51, Sycp3 and Sycp1. Non-irradiated siblings of each genotype were also processed in the same manner as controls. We observed a significant increase of Dmc1/Rad51 signals (average of total area stained by Dmc1/Rad51 foci per nucleus) in irradiated *spo11*−/− spermatocytes (0.50±0.64) compared to non-irradiated control spermatocytes (1.39±1.02), although Dmc1/Rad51 signals were not completely recovered to wild-type leptotene or early zygotene-levels. Notably, we observed a similar increase of Dmc1/Rad51 signal in irradiated *sycp2*−/− spermatocytes (0.52±0.54) compared to non-irradiated control spermatocytes (1.25±0.68). Since formation of Dmc1/Rad51 foci was partly recovered by exogeneous induction of DSBs, this result supports the idea that Sycp2 is involved in meiotic DSB formation.

We also examined the formation of RPA foci in *sycp2*−/− spermatocytes by immunostaining spermatocyte spreads with an anti-human RPA antibody (Fig 7A). RPA is a single-stranded DNA-binding protein that is required for the recruitment of DMC1/RAD51 in mice [39]. In wild-type spermatocytes, RPA foci appeared at leptonema (Fig 7A-i), became bright foci or short stretches in zygonema (Fig 7A-ii and 7iii), and disappeared in pachynema (Fig 7A-iv). However, similar to Dmc1/Rad51 foci, RPA foci were rarely detected in *sycp2*−/− spermatocytes (Fig 7A-v and 7A-vi).

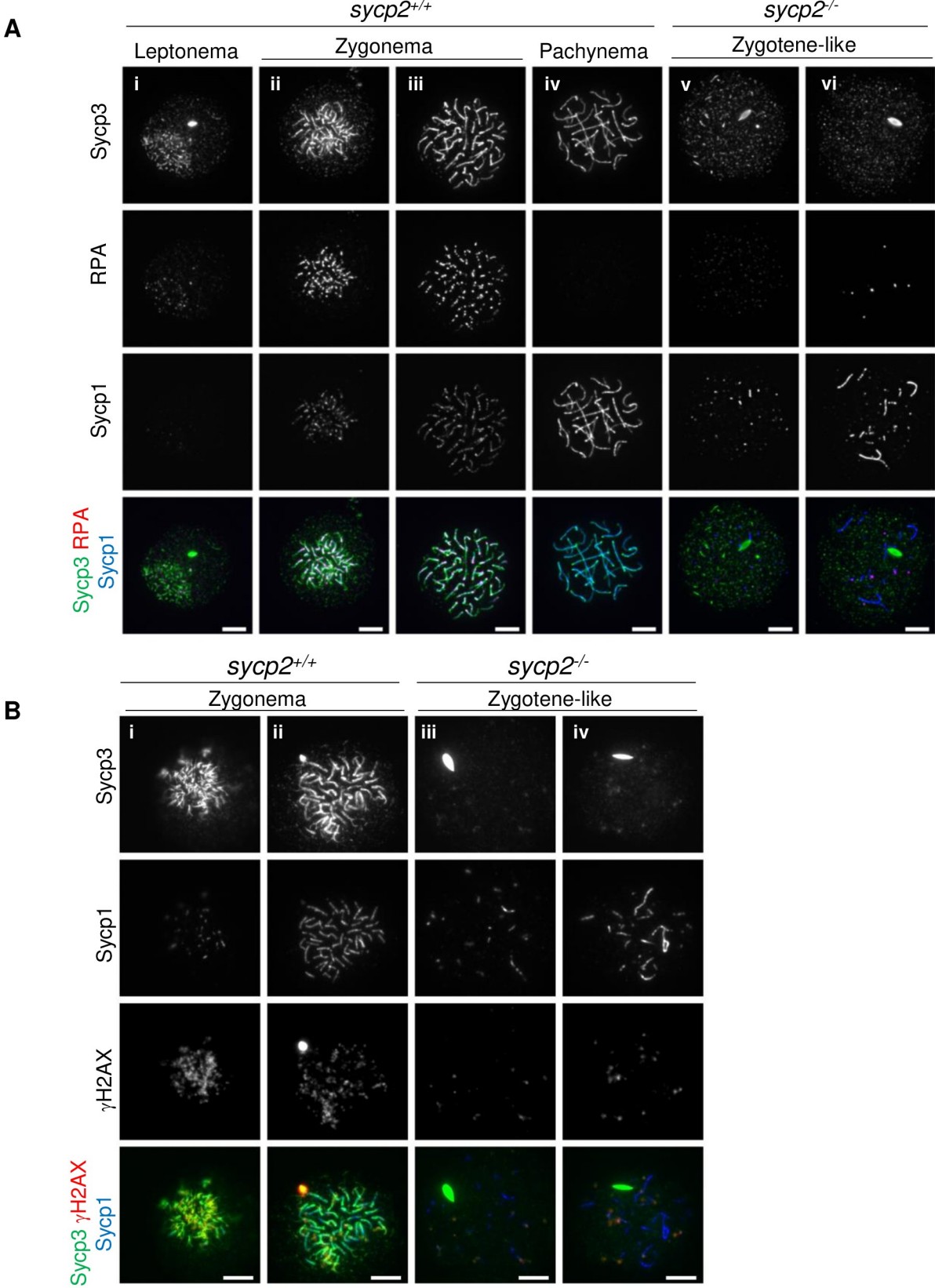

**Fig 7. RPA and γH2AX signals were severely reduced in *sycp2*⁻/⁻ spermatocytes.** A: Staining of *sycp2*⁺/⁺ and *sycp2*⁻/⁻ spermatocyte chromosomal spreads with anti-human RPA, anti-Sycp1 and anti-Sycp3 antibodies. Note that the human RPA protein (NP_002936.1) shares

63.5% identity and 80.9% similarity with the zebrafish protein (NP_956105.2). In $sycp2^{+/+}$ spermatocytes, RPA appears at leptonema in faint foci, which become brighter at zygonema and disappear at pachynema. In $sycp2^{-/-}$ spermatocytes, RPA foci were rarely detected; only a few foci were observed in some cells. B: Immunostaining of $sycp2^{+/+}$ (i, ii) and $sycp2^{-/-}$ (iii, iv) spermatocyte chromosomal spreads with anti-γH2AX, anti-Sycp3 and Sycp1 antibodies. Individual images with each antibody and a merged image are shown. The wild-type ($sycp2^{+/+}$) nuclei are at early to mid-zygonema. Scale bars, 5 μm.

We further examined the phosphorylation of the histone H2A variant H2AX (γH2AX), which occurs in response to DNA DSBs, in wild-type and $sycp2^{-/-}$ spermatocytes (Fig 7B). In wild-type spermatocytes, γH2AX signals appeared from leptonema to early zygonema (Fig 7B-i) and diminished towards pachynema (Fig 7B-ii). The signals were observed as scattered patches, mostly in subnuclear regions where Sycp3 fragments clustered. This observation is consistent with the localization of Dmc1/Rad51 (S9 Fig) and Rad51 [26] on Sycp3-stained axes. Next, we compared γH2AX signals in $sycp2^{-/-}$ spermatocytes that exhibited short Sycp1 fragments with those in wild-type spermatocytes at early zygonema. Strikingly, the γH2AX signals were drastically reduced in $sycp2^{-/-}$ spermatocytes compared to wild-type spermatocytes (Fig 7B-iii). Some $sycp2^{-/-}$ spermatocytes were stained with γH2AX in patches spread over nuclei rather than as clusters in a subnuclear domain.

Taken together, these results indicate that the early steps of meiotic recombination are defective in the absence of Sycp2.

## Discussion

### Sycp2 is required for the ovarian development in zebrafish

In zebrafish, all juveniles develop gonads with immature oocytes regardless of their definitive sex, and individuals in which these immature oocytes degenerate become males [29–31], and oocyte-depleted mutants develop as males that are mostly infertile [32–35]. For instance, zebrafish homozygous for mutations in Piwi-interacting RNA pathway genes, such as *ziwi* (zebrafish *piwil1*) and *hen1*, are all phenotypically males [32,40]. Zebrafish with homozygous mutations in the *fancl* and *brca2* genes, which code DNA repair proteins, also fail to develop ovaries during sexual differentiation and become sterile males [36,41]. In this study, we observed similar male bias in $sycp2^{-/-}$ fish (Table 1). This implies that oocytes could be eliminated during gonad development in $sycp2^{-/-}$ fish. Notably, such male bias is not observed in zebrafish homozygous for *spo11* mutations, and mutant females generate eggs capable of fertilization (Table 1) [26]. Interestingly, in mice, both *Sycp2* and *Spo11* mutant females show less severe phenotypes than those in mutant males [22,42]. In addition, we observed previtellogenic dictyate oocytes in four of the five *sycp2* mutants at 28 days postfertilization (S4 Fig). Oocytes of *sycp2* mutants seem to develop slightly later than those of *fancl* and *brca2* mutants [36,41]. Meiotic checkpoint and/or cell elimination mechanisms may be different between mice and zebrafish and among the mutated genes, as well as between sexes. The checkpoint may be rather active in oogenesis in *sycp2* mutant zebrafish than in *spo11* mutants. Since p53-mediated germ cell apoptosis is known to induce sex reversal by compromising the survival of zebrafish oocytes [36], a future interest will be to identify differences in such checkpoint activation among these meiotic mutant zebrafish.

### SC assembly in zebrafish spermatocytes

In this study, we developed anti-Sycp2 antibodies to understand the spatiotemporal processes of all SC components in wild-type and *sycp2* mutant spermatocytes (see a model in S12 Fig). Interestingly, Sycp2 appeared to be essential for the integration of Sycp3 into axial elements,

since Sycp3 persistently formed aggregates in *sycp2* mutant spermatocytes that were reminiscent of those observed in wild-type preleptonema, when Sycp2 had yet to form the axis (Fig 3). In mice, SYCP2 and SYCP3 form antiparallel heterotetramers [19], and the interaction of the C-terminal coiled-coil domains of SYCP2 and SYCP3 is required for the axis association of SYCP3 [22]. Thus, the SYCP2-SYCP3 interaction seems to play a conserved role in the formation of axial elements in vertebrates. We observed abnormal formation and localization of Sycp1 filaments in *sycp2* mutant spermatocytes (Fig 3 and Fig 4), indicating that Sycp2 is also important for proper loading of Sycp1 onto meiotic chromatin in zebrafish, as reported in *Sycp2* mutant mice [22]. This phenotype was different from *spo11* knockout zebrafish, in which Sycp1 filaments were not observed in the majority of spermatocytes [26]. The colocalization of Sycp1 filaments and paired BAC foci was observed in a few *sycp2*$^{-/-}$ spermatocytes (Fig 6A-iv and 6B). Therefore, at least some aberrant Sycp1 filaments seem to localize on such partly paired regions of homologous chromosomes in the absence of Sycp2. We also observed costaining of aberrant Sycp1 filaments in *sycp2*$^{-/-}$ spermatocytes with stretches of Rad21l1 that seem to exist, at least in part, even in the absence of Sycp2 (S8 Fig). It was difficult to determine whether these Sycp1 filaments were loaded between sister chromatids, at our image resolution. In the case that the aberrant Sycp1 filaments were formed between sister chromatids, as observed in a hypomorphic *Stag3* mutant mouse, in which the levels of the cohesin subunit REC8 are partly reduced [43], Sycp2 might function to prevent formation of the synaptonemal complex between sister chromatids.

## Sycp2 plays a critical role in early meiotic recombination and homologous pairing

Meiotic chromosomal axes have been thought to serve as scaffolds for meiotic DSB proteins from observations in *Saccharomyces cerevisiae*. In *S. cerevisiae*, mutations of the axis proteins Hop1 and Red1 severely impair the formation and repair of meiotic DSBs [2–6]. In addition, Red1 functions in homologous pairing, likely by potentiating Hop1-promoted pairing [44,45]. In mammals, HORMAD1 is likely to be a counterpart of budding yeast Hop1 since it localizes on the axis prior to DSB formation and since meiotic DSBs are reduced in *Hormad1* knockout mice [10,46–48]. The recent work of West et al. noted that Red1 and SYCP2/SYCP3 share conserved biochemical properties as axis core proteins and that an N-terminal domain of mouse SYCP2 binds to the HORMA domain [19]. The putative HORMA-binding domain is also conserved at the sequence level in zebrafish Sycp2 (S13 Fig). In this study, we generated *sycp2* null mutant zebrafish and showed that Sycp2 is required for Dmc1/Rad51 and RPA focus formation and γH2AX signals during meiotic prophase I (Fig 6 and Fig 7). A partial recovery of Dmc1/Rad51 focus formation after induction of exogenous DSBs implies that Sycp2 is required for meiotic DSB formation (S11 Fig). Furthermore, chromosome pairing at telomeres and a homologous locus were severely impaired in the absence of Sycp2 (Fig 4 and Fig 5). Our observation that DSB marker signals and homologous pairing were largely diminished in *sycp2* mutant zebrafish confirmed *in vivo* that Sycp2 has critical functions in early meiotic recombination, as reported for budding yeast Red1. Sycp2 is a large protein that could potentially work as a scaffold for meiotic proteins. Further identification of Sycp2 interactors will provide information to understand how Sycp2 coordinates multiple meiotic events, such as telomere clustering, DSB formation and repair, and synapsis.

In this study, we showed the localization of Dmc1/Rad51 foci near telomeres in zebrafish spermatocytes (Fig 6, S9 Fig). This observation is consistent with previous observations of Rad51 localization, indicating that DSB formation occurs near telomeres in zebrafish spermatocytes [26,49]. Interestingly, meiotic DSBs occur frequently in human males at genomic regions close to telomeres [50]. In many mammals, including rodents and primates, the

meiosis-specific histone methyltransferase PRDM9 is known as a major determinant of hot-spots [51–55]. Since PRDM9 binds to a specific DNA motif, its DNA-binding property is thought to play a key role in hotspot determination. Since zebrafish do not have a functional *prdm9* ortholog [56], it is possible that the telomere-proximal bias of DSBs rather than a PRDM9-mediated mechanism primarily underlies hotspots. Our observations that DSB signals were largely diminished in the absence of Sycp2 (Fig 6 and Fig 7) suggest that the formation of the axis structure could play a key role in the regulation of meiotic DSBs. Therefore, zebrafish will be an ideal model with which to approach this new aspect of DSB control based on chromatin structures at subtelomeric regions in human males.

## Materials and methods

### Ethics statement

All experiments were conducted in accordance with Rules for Animal Experiments at the National Institute of Genetics, Research Organization of Information and Systems. The experimental plan using zebrafish is approved by the National Institute of Genetics official ethics committee (Approval Number 27–12, 28–13, 29–13, 30–14, 31–18).

### Fish

Zebrafish (*Danio rerio*) were maintained under standard conditions as described in The Zebrafish Book [57]. The wild-type *India* line (see ZFIN at http://zfin.org/action/genotype/view/ZDB-GENO-980210-28) has been maintained in our laboratory for more than 20 generations. The AB* line (see ZFIN at http://zfin.org/action/genotype/view/ZDB-GENO-960809-7) was kindly provided by Prof. U. Strähle in 2010 and has been maintained in our laboratory. The *its* fish have been described previously: *its* mutant fish were derived from the wild-type *Tübingen* line and then crossed with the IM strain (wild-type inbred) [58] to obtain heterozygous mutant fish [25].

Knockout lines of *sycp2* were generated by TALEN mutagenesis targeting exon 18 with the following sequences: 5'-TGATGAAGCCAGCTCT-3' (left TALEN) and 5'-GTGTCTCTCC TTCTTT-3' (right TALEN). TALEN modules targeting these sequences were cloned into pCS2TAL3DD and pCS2TAL3RR vectors using the Golden Gate assembly method [59,60]. TALEN mRNAs were prepared with an mMESSAGE mMACHINE SP6 kit (Ambion). Injection with ~1 ng of mRNA was performed on wild-type *India* embryos at the 1- to 4-cell stages. Founder males were mated with *India* females, and the F$_1$ siblings were screened by genotyping. Homozygous *sycp2* knockout fish were obtained by mating F$_1$ siblings carrying either 14-bp or 16-bp deletions at the TALEN target site (S1 Fig).

The *spo11* knockout line was generated by CRISPR-Cas9 mutagenesis based on published protocols [61,62]. Template DNA for single-guide RNA (sgRNA) synthesis was prepared by amplification with a primer specific to *spo11* exon 5, a universal reverse primer (S3 Table) and T4 DNA polymerase. After purification of the template DNA, sgRNA was transcribed *in vitro* with a MEGAscript T7 kit (Ambion) and purified with a MEGA clean-up kit (Ambion). Wild-type AB* embryos were injected at the 1- or 2-cell stage with 2.3 nl of a mixture of 10 pmol/μl Cas9 NLS protein (abm) and 100 ng/μl *spo11* sgRNA. Founders were backcrossed with AB* fish, and the F$_1$ siblings were screened by genotyping. Homozygous *spo11* knockout fish were obtained by mating F$_1$ siblings carrying a +1 frameshift mutation in exon 5 (S10 Fig).

### Genotyping and mapping of the *its* mutation

The fish were genotyped after extracting genomic DNA from caudal fin clips. PCR was performed with GoTaq Green Master Mix (Promega) using specific primers for each site (S3

Table). The PCR products were examined by either agarose or acrylamide gel electrophoresis. Sequencing of the genomic *sycp2* locus was performed by amplifying exons 8 to 11 of *sycp2* with KOD -Multi & Epi- (Toyobo). Some PCR products were treated with ExoSAP-IT PCR Product Cleanup Reagent and were analyzed by general Sanger sequencing. Other PCR products were cloned into the pGEM-T vector and then sequenced with T7 or SP6 primers.

For mapping of the *its* mutation, 682 sterile *its* mutant fish were genotyped as described above using primers to detect SSLPs between the *Tübingen* and the IM lines, and the causal genomic region was mapped between the z20895 and z7550 SSLP markers.

### Fertilization tests

To assess the fertility of *sycp2$^{-/-}$* zebrafish, six *sycp2$^{-/-}$* and four wild-type siblings were individually mated with wild-type females. For the complementation test, five *sycp2$^{its/-}$* siblings, one *sycp2$^{IM/-}$* sibling and one wild-type sibling were individually mated with wild-type females. After the eggs were collected, all eggs were incubated at 28°C for 6 hours, and the gastrulae and unfertilized eggs were counted.

### Evaluation of sexual phenotypes

To examine sexual phenotypes of *sycp2$^{+/-}$* or *spo11$^{+/-}$* mutant fish, offspring were obtained from mass mating of heterozygote fish. Offspring fish were dissected at 8 to 9-week old, and the morphology of the gonads was examined with a Nikon ECLIPSE TE200-S microscope. To facilitate the observation of gonad morphology, dissected gonads were placed on a slide with a drop of Leibovitz L-15 media (Sigma) and flattened with a coverslip. Gonads were classified into either ovaries with oocytes or testes with lobule structures and/or with spermatocytes, spermatids, or sperm cells (S14 Fig). Fin clips of offspring fish were used for genotyping.

### Preparation of testis cDNA and RT-PCR

Total RNA was extracted from fin clips or paired testes with RNAiso Plus (Takara) and treated with TURBO DNase (Ambion). Testis cDNA was prepared from 1 μg of total RNA with a PrimeScript RT-PCR kit (Takara) using oligo dT primers. qPCR was performed with LightCycler 480 SYBR Green I Master on a LightCycler 480 system (Roche) with the default settings, except that annealing was performed at 58°C. For the cloning of *sycp2* cDNA and the analysis of *sycp2* splicing at exons 8–9, PCR was performed using KOD -Multi & Epi- (Toyobo) on cDNA from five testes each of *sycp2$^{+/+}$* and *sycp2$^{its/its}$* males. All primers used in this study are listed in S3 Table.

### Histological analysis

Adult testes and bodies of juvenile fish at 28 days postfertilization were fixed in Bouin's solution and 4% paraformaldehyde in PBS, respectively, at 4°C overnight. The fixed samples were dehydrated in an ethanol series (70%, 80%, 90% and 100%), methyl benzoate, and Lemosol (Wako) and embedded in paraffin. Hematoxylin-eosin (HE) staining was performed with 5-μm-thick sections.

### Mini-gene splicing assay

Two mini-gene constructs containing either wild-type or *its sycp2* intron 8 with flanking exon 8 and 9 sequences were constructed (S3 Fig). The genomic *sycp2* sequence spanning exons 8 to 9 was amplified from wild-type and *its* homozygote genomic DNA by PCR using primers with restriction sites (S3 Table). The PCR products were cloned into pT2AL200R150G [63] using

HindIII and BamHI sites. The purified plasmids and transposase mRNA were injected into wild-type embryos at the 1- to 4-cell stages as previously reported [63]. After injection, the expression of each mini-gene construct was verified by fluorescence signals and by RT-PCR of the reporter EGFP (S3 Fig).

## Antibodies

The antibodies used in this study are listed in S2 Table. Polyclonal antibodies specific for zebrafish Sycp2,Dmc1/Rad51 and Rad21l1 were generated in this study (S5 Fig, S8 Fig). *sycp2* cDNA coding amino acid residues 1071–1569 (C499) or 1238–1569 (C332) was cloned into the pET-21a vector. The recombinant Sycp2 proteins were expressed in *Escherichia coli* Rosetta-gami 2 (DE3) and purified by Ni-NTA agarose (Qiagen). The purified proteins were separated on a 10% acrylamide gel, and a band corresponding to each Sycp2 recombinant protein was cut out after Coomassie brilliant blue staining. The gel sections were washed in water and homogenized in PBS. After isolation of the Sycp2 recombinant proteins, 500 μg each of the recombinant proteins C499 and C332 was used to immunize guinea pigs and rats, respectively (Evebioscience). After the fifth injection of antigen, guinea pig and rat anti-Sycp2 antisera were recovered and affinity-purified using CNBr Sepharose beads (GE Healthcare) conjugated with the recombinant Sycp2 protein C499. To generate anti-Dmc1 antiserum, a Dmc1 cDNA coding amino acid residues 8–220 (NP_001018618.1) was cloned into the pQE-30 vector (Qiagen). The recombinant Dmc1 protein was expressed in *E. coli* M13 and purified by Ni-NTA agarose (Qiagen). The purified Dmc1 proteins were used to immunize guinea pigs (BioGate). Unpurified anti-Dmc1 antiserum was used in this study. It should be noted that the protein sequence of the Dmc1 antigen is 43% identical to the zebrafish Rad51 protein. Thus, our anti-Dmc1 antiserum potentially recognizes Rad51. An anti-Rad21l1 antiserum was generated by immunizing mice with the full-length zebrafish Rad21l1 protein. The full-length cDNA of Rad21l1 was amplified from AB* testis cDNAs and cloned into the pET-28a vector using the NEBuilder HiFi DNA Assembly (NEB).

## Preparation of spermatocyte chromosomal spreads

Spermatocyte chromosomal spreads were prepared from adult testes by a dry-down method [64] adapted for zebrafish. Briefly, a pair of testes was dissected from anesthetized fish and cut into small pieces in PBS. The cells were dissociated by gentle pipetting, and the cell suspension was transferred into a new tube, leaving behind any nondissociated clumps. After washing in PBS, the cells were incubated for 8 min in hypotonic buffer (30 mM Tris-HCl pH 8.2, 50 mM sucrose, 17 mM sodium citrate, 5 mM EDTA, 0.5 mM dithiothreitol, 1x cOmplete Protease Inhibitor Cocktail (Roche)). After washing in PBS, the nuclear pellets were recovered in PBS. The nuclear suspension was mixed with a 2x volume of 100 mM sucrose, and ~$10^5$ nuclei were spread on a slide dipped in a solution of PBS, 1% paraformaldehyde, and 0.05% Triton X-100. The spread slides were incubated in a humid chamber for 1 hour and then half-dried. After washing twice in 0.1% Tween 20, the slides were kept at -80˚C or immediately used for immunostaining.

## Immunostaining

Blocking of the spread slides was performed with a solution of PBS, 5% skimmed milk, and 5% goat and/or donkey serum. The slides were incubated at room temperature overnight with primary antibodies and at 37˚C for 1 hour with secondary antibodies. All antibodies and their dilutions at the time of use are listed in S4 Table. After the last washing step in PBS containing 10 ng/ml DAPI, the slides were mounted using VECTASHIELD Antifade Mounting Medium

(Vector Labs). For telomere staining, TEN buffer (100 mM NaCl, 20 mM Tris-HCl pH 7.4, 1 mM EDTA) was used instead of PBS, and incubation with secondary antibodies was performed with 0.5 µg/ml DAPI and 15 nM TRed-HPTH59-A (telomere-targeting polyamide; HiPep Laboratories) [37] at room temperature for 3–4 hours. Note that TE buffer (20 mM Tris-HCl pH 7.4, 1 mM EDTA) was used for the wash step before incubation with the secondary antibody/DAPI/telomere-targeting polyamide mix, as salt prevents binding of the telomere-targeting polyamide.

## Bacterial artificial chromosome (BAC) probe staining

BAC probe DNA and the Cot-1 DNA were prepared as previously described using BAC clone CH211-31P3 and salmon sperm DNA (Wako), respectively [26]. The BAC probe staining protocol was adapted from [26]. First, immunostaining of chromosomal spread was performed as described above. After immunostaining, the slides were postfixed with 400 µl of 1% paraformaldehyde in PBS for 15 min at room temperature. The slides were then washed in PBS and dehydrated by placing in 70%, 85% and twice in 100% ethanol. Dried slides were incubated with 10 µl of probe mixed with 25 µg of Cot-1 at 73˚C for 3 min. After coming to temperature at ~40˚C, slides were incubated in a humid chamber at 37˚C overnight. Washing steps were performed as described previously [26]. The distance between BAC foci was quantified as previously described [26]. Two foci with an intervening distance of <3 µm were considered to be paired.

## Western blotting of Sycp2

Testes from 8–14 fish were homogenized in 5 ml of RIPA buffer. After freezing at -80˚C and sonication, the soluble fraction was recovered by centrifugation at 20,000 x g for 15 min at 4˚C. The concentrations of the protein extracts were determined with a Protein Assay Lowry Kit (Nacalai). Immunoprecipitation was performed from ~2.5 mg of testis proteins of each genotype with either 25 µg of guinea pig anti-Sycp2 antibody or 25 µg of normal guinea pig IgG. The antigen-antibody complex was recovered with 50 µµl of protein A Sepharose beads (GE Healthcare). The immunoprecipitated samples were eluted in 40 µl of 2x sample buffer at 95˚C for 10 min and migrated on a 7.5% Mini-PROTEAN TGX precast protein gel (Bio-Rad) at 200 V for 30 min. After transfer to an Immobilon-P membrane (Millipore) at 100 V for 70 min, immunoblotting was performed with a rat anti-Sycp2 antibody and a goat anti-rat HRP-conjugated antibody (Santacruz) using Can Get Signal Immunoreaction Enhancer Solution (Toyobo). The signals were developed with Western BLoT Ultra Sensitive HRP Substrate (Takara) and captured with an ImageQuant LAS 4000 Mini (GE Healthcare).

## Image analysis and quantification

Histological images were captured with an Olympus BX51 microscope equipped with a Keyence VB7010 camera. Images of the embryos were captured with a Leica MX16 FA fluorescence microscope equipped with a Leica DFC310 FX camera. Cytological images were captured with a DeltaVision PersonalDV-TM fluorescence microscope with softWoRx software (GE Healthcare). All cytological images were processed using OMERO (OME) [65] and Fiji [66]. For telomere counting (Fig 4) and Dmc1/Rad51 imaging (Fig 6, S9 Fig and S10 Fig), stacks of 10 images along the z-axis (section spacing: 0.2 µm) were deconvoluted with softWoRx and projected with Fiji. The Sycp1 fragments and their colocalization with telomere foci stained by the telomere-targeting polyamide were manually counted in each nucleus. Punctate Sycp1 signals were not counted as Sycp1 fragments. To calculate the Dmc1/Rad51 focus-stained area from binary images, the same threshold was applied to all images in the

same experiment to eliminate background signals. All quantifications were performed by a Fiji macro with manually selected DAPI-positive ROIs (available upon request).

### γ-ray irradiation of zebrafish males

Adult zebrafish were placed in a plastic box, with water depth of 2 cm. Animals were exposed to γ-rays for 11 minutes 42 seconds at 0.855 Gy/min (total exposure 10 Gy) in a Cs-137 Gammacell 40 Exactor (MDS Nordion). Irradiated fish were anesthetized 30 minutes after irradiation, and spermatocyte chromosomal spreads were prepared as indicated above. For each genotype, non-irradiated siblings were processed as controls.

## Supporting information

**S1 Fig. Generation of *sycp2* knockout fish by TALEN mutagenesis.** A: A schematic presentation of the exon-intron structure of the *sycp2* gene. The Sycp2 protein is coded in 45 exons (shown as vertical lines) that span ~60 kbp on zebrafish chromosome 23. The coding sequences (exons) are based on the annotations for XM_679956.6. B: Mutation sites of *sycp2* knockout zebrafish generated by TALEN mutagenesis. The sequences targeted by the TALEN proteins are shown in yellow. Three *sycp2* mutant lines with 3-, 14- or 16-bp deletions were isolated. These alleles were named *sycp2^Δ3^*, *sycp2^Δ14^* and *sycp2^Δ16^*, respectively.
(TIF)

**S2 Fig. Sequences of *sycp2* cDNA obtained from *sycp2^its/its^* testes.** A: A sequence of the full-length wild-type *sycp2* cDNA. B: A sequence of the full-length *sycp2* cDNA with the insertion of a premature termination codon (in red) by aberrant exon 8–9 splicing. The exon 8 and exon 9 sequences are boxed. Both wild-type (A) and *its*-type (B) cDNAs were obtained from the same *sycp2^its/its^* male fish.
(TIF)

**S3 Fig. Mini-gene splicing assay.** A: A schematic presentation of the mini-gene constructs used in the splicing assay. The mini-genes contained either wild-type (WT) or *its*-type (*its*) sequences for *sycp2* intron 8 (133 bp) and flanking regions of exon 8 (34 bp) and exon 9 (143 bp). The primers used for RT-PCR to amplify a control GFP sequence (S3 Fig) and to assess exon 8–9 splicing (Fig 2D) are shown as green and red arrows, respectively. Tol, Tol2 transposon sequences. $P_{EF1\alpha}$, elongation factor 1α promoter derived from *Xenopus laevis* for ubiquitous expression of the transgene. pA, poly(A) signal. Kozak, Kozak consensus sequence. The size of each element does not correspond to its actual sequence length. B: Expression of GFP in transgenic embryos with wild-type (WT E8-9) and *its*-type (*its* E8-9) mini-genes. Bright-field and fluorescent (GFP) images of embryos at one day postfertilization. The fluorescent signals indicate mini-gene expression. Scale bar, 300 μm. C: Mini-gene splicing assay in the wild-type genetic background. In addition to the clones shown in Fig 2D, RT-PCR was performed with two more transgenic fish each with either wild-type (WT E8-9) or *its*-type (*its* E8-9) *sycp2* mini-genes. One more wild-type fish without mini-genes was used as a control (-). D: Expression of the GFP reporter gene in transgenic fish with mini-genes. RT-PCR was performed with caudal fin cDNA from two individual wild-type fish without a transgene (-) and three individual transgenic fish with either wild-type (WT E8-9) or *its*-type (*its* E8-9) *sycp2* mini-genes. The clones labeled "1" and those labeled "2" and "3" were used for RT-PCR in Fig 2D and S3 Fig, respectively. Non-RT (-) controls were subjected to PCR using total RNA processed without an RT reaction.
(TIF)

**S4 Fig. Histology of *sycp2*<sup>+/+</sup> and *sycp2*<sup>-/-</sup> juvenile gonads.** HE-stained gonads of two *sycp2*<sup>+/+</sup> (WT-1 and -2) and five *sycp2*<sup>-/-</sup> (KO-1 to -5) zebrafish at 28 days postfertilization. Arrows indicate late stage IB oocytes [36]. Scale bars, 20 μm.
(TIF)

**S5 Fig. Generation of anti-Sycp2 antibodies and anti-Dmc1 antiserum.** A: A schematic model of the zebrafish Sycp2 protein. The full-length structure of the 1569-amino-acid sequence is shown, with regions similar to mammalian SYCP2 domains: NTD, N-terminal domain; HORMA-BD, putative HORMA-binding domain; CC, C-terminal coiled-coil domain [19]. The C-terminal regions used as immunogens to generate anti-Sycp2 antibodies (C499 and C322) are indicated as blue bars. The positions of premature stop codon in *sycp2* mutant lines are indicated with red arrows. B: Western blotting of *sycp2*<sup>+/+</sup>, *sycp2*<sup>-/-</sup> and *sycp2*<sup>its/its</sup> testis protein extracts using an anti-Sycp2 antibody. Immunoprecipitation was performed with protein extracts from *sycp2*<sup>+/+</sup>, *sycp2*<sup>-/-</sup> and *sycp2*<sup>its/its</sup> testes using a guinea pig anti-Sycp2 antibody (IP-Sycp2) or normal guinea pig IgG as a control (IP-IgG). SDS-PAGE was performed on 7.5% TGX Precast Gel (Bio-Rad). Each well was loaded with an immunoprecipitated sample or 0.4% input. Immunoblotting was performed with a rat anti-Sycp2 antibody. The predicted size of Sycp2 is 176 kDa. The left part is a colorimetric image of the protein ladder on the same membrane. The Sycp2 protein was not detected in *sycp2*<sup>its/its</sup> testes. However, we cannot exclude the possibility that there is expression of a truncated Sycp2 protein that is not recognized by our anti-Sycp2 antibodies specific to a C-terminal region of Sycp2. C: Western blotting of wild-type testis extract with anti-Dmc1 guinea pig antiserum. Each well was loaded with 28 μg of protein. The predicted size of zebrafish Dmc1 is 38 kDa. After blocking in TBST with 5% skimmed milk, the membrane was incubated with the anti-Dmc1 guinea pig antiserum at a 1:2500 dilution and with a biotinylated anti-guinea pig antibody at 1:1000; then, the signals were amplified with a VECTASTAIN ABC kit (Vector Labs) and developed with an ECL Plus kit (lane -). Blotting was also performed with anti-Dmc1 antiserum after absorption to recombinant Dmc1 proteins (lane +) as a control. The left image is a colorimetric image of the protein ladder on the same membrane.
(TIF)

**S6 Fig. Quantification of Sycp1 filaments and immunostaining of SC components in *sycp2*<sup>Δ16/Δ16</sup> spermatocytes.** A: Quantification of the number of Sycp1 fragments per nucleus in *sycp2*<sup>+/+</sup>, *sycp2*<sup>its/its</sup> and *sycp2*<sup>-/-</sup> (*sycp2*<sup>Δ14/Δ14</sup>) spermatocytes. Quantification was performed for nuclei stained with an anti-Sycp1 antibody, a telomere-targeting polyamide and DAPI (see Materials and Methods). Since zebrafish have 25 pairs of homologous chromosomes with telomeres at both ends, nuclei containing ~50 and 25 Sycp1 fragments are at zygonema and pachynema, respectively (Z and P). Wild-type nuclei containing between 25 and 50 Sycp1 fragments are likely at the zygotene-pachytene transition (Z-P). The boxes indicate the SD, with the mean value as the middle bar. Chromosomal spreads of one (wild-type) or two (*sycp2*<sup>its/its</sup> and *sycp2*<sup>-/-</sup>) individual fish were used for counting. *sycp2*<sup>+/+</sup>, n = 34; *sycp2*<sup>its/its</sup>, n = 41; and *sycp2*<sup>-/-</sup>, n = 45. *** indicates $p < 0.0001$ (Student's t-test). B: Immunostaining of SC components on *sycp2*<sup>+/+</sup> and *sycp2*<sup>Δ16/Δ16</sup> spermatocyte chromosomal spreads. Individual images with anti-Sycp3, anti-Sycp2, or anti-Sycp1 antibodies and a merged image are shown for each nucleus. The white line on the wild-type leptotene image indicates a nuclear border with another nucleus on the top left. Scale bars, 5 μm.
(TIF)

**S7 Fig. Staining of Sycp3, telomeres and Sycp1 on wild-type spermatocyte chromosomal spreads.** A. Individual images with telomere-targeting polyamide, anti-Sycp3 antibodies or

anti-Sycp1 antibodies and a merged image are shown for each nucleus. The stages of the nuclei were determined based on the Sycp3 and Sycp1 signals, according to a recent report by Blokhina et al. [26]: in preleptonema, telomeres cluster together to form a bouquet (i); this step is followed by the extension of Sycp3 filaments from the clustered telomeres at leptonema (ii); in zygonema, following the inward extension of the Sycp3 filaments, Sycp1 exclusively emanates from telomeres that disperse throughout the nuclei, resulting in the dismantling of the telomere bouquet; Sycp1 and Sycp3 filaments extend along the entire lengths of chromosomes emanating from both telomeres in pachynema (vi). B: Costaining of telomeres and Sycp1 on $sycp2^{+/+}$ and $sycp2^{\Delta16/\Delta16}$ spermatocyte chromosomal spreads. The regions outlined in white are shown at a higher magnification at the bottom. The nuclei of $sycp2^{+/+}$ spermatocytes are at zygonema (left; telomeres detected at one end of each Sycp1 filament) and pachynema (right; telomeres detected at both ends of Sycp1 filaments). Scale bars, 5 μm.
(TIF)

**S8 Fig. Immunostaining of Rad21l1, Sycp1 and Sycp3 on wild-type and $sycp2^{-/-}$ spermatocyte chromosomal spreads.** A: The meiotic cohesin Rad21-like 1 (Rad21l1) was costained with Sycp1 and Sycp3. In wild-type spermatocytes, Rad21l1 showed similar localization to Sycp3 throughout meiotic prophase I, although scattered signals were also observed over nuclei. In $sycp2^{-/-}$ spermatocytes, Rad21l1 was observed as mottled stretches on Sycp1 filaments as well as scattered signals over nuclei. Arrowheads indicate Rad21l1 costained with Sycp1 filaments in $sycp2^{-/-}$ spermatocytes. Scale bars, 5 μm. B: Western blot of zebrafish testis extract with the anti-Rad21l1 antiserum used in S8A Fig. Adult testis proteins were extracted in RIPA buffer, and 50 μg of protein was migrated on a 10% SuperCep acrylamide gel (Wako). After blocking in TBST with 5% skimmed milk, the membrane was incubated with the anti-Rad21l1 mouse antiserum at a 1:150 dilution and with anti-mouse IgG HRP-conjugated antibody at 1:2000; then, the signals were detected with an ECL Prime kit. The right part of the image is a colorimetric image of the protein ladder on the same membrane. Although a single band appeared larger than the predicted molecular weight of zebrafish Rad21l1 (63 kDa), such difference has been observed for mouse RAD21L [67].
(TIF)

**S9 Fig. Staining of Sycp3, telomeres and Dmc1/Rad51 on wild-type spermatocyte chromosomal spreads.** Individual images with anti-Sycp3 antibody and merged images are shown for two wild-type nuclei at leptonema to early zygonema. The merged images show staining for Sycp3 (blue), telomeres (magenta), Dmc1/Rad51 (green) and DAPI (white). The regions enclosed by white rectangles are shown in a higher magnification with or without Sycp3 (images in the second and third rows). Scale bars, 5 μm.
(TIF)

**S10 Fig. Generation of a $spo11$ knockout line by CRISPR-Cas9 mutagenesis.** A: A schematic presentation of the exon-intron structure of the $spo11$ gene. We isolated a mutant with a +1 frameshift caused by substitution of "CA" with "ATT" in exon 5. In this study, this mutation is referred to as the $spo11^{-}$ allele. The sequencing data for the mutation site in $spo11^{+/+}$ (wild-type) and $spo11^{-/-}$ are shown. B: Staining of γH2AX on $spo11^{+/+}$ and $spo11^{-/-}$ spermatocyte chromosomal spreads. In wild-type spermatocytes, γH2AX signals were detected at leptonema to early zygonema, as we reported previously [25]. In contrast, γH2AX signals were rarely detected in $spo11^{-/-}$ spermatocytes with Sycp3 patterns similar to those in wild-type leptonema or early zygonema. Thus, there were no detectable DSBs in our $spo11^{-/-}$ zebrafish spermatocytes as determined by γH2AX staining. Scale bars, 5 μm.
(TIF)

**S11 Fig. Dmc1/Rad51 focus formation in wild-type, *spo11*[-/-] and *sycp2*[-/-] spermatocytes after γ-ray irradiation.** A: Immunostaining of Dmc1/Rad51, Sycp1 and Sycp3 on non-irradiated (no irradiation; i to vi) and γ-ray irradiated (10Gy; vii to xii) spermatocyte chromosomal spreads of wild-type (i to iv, vii to x), *spo11*[-/-] (v and xi) and *sycp2*[-/-] (vi and xii). The wild-type nuclei are at leptonema (i and vii), early zygonema (ii and viii), late zygonema (iii and ix) and pachynema (iv and x) according to the Sycp1 and Sycp3 staining patterns. *spo11*[-/-] nuclei at a leptotene- or early zygotene-like stage (L/EZ-like), according to Sycp3 staining patterns, are shown (v and xi). Early zygotene-like (EZ-like) *sycp2*[-/-] nuclei stained with short Sycp1 fragments are shown (vi and xii). The white line on the irradiated *sycp2*[-/-] image (xii) indicates a nuclear border with another nucleus on the top left. Scale bars, 5 μm. B: Quantification of the Dmc1/Rad51-stained area in non-irradiated (-) and γ-ray irradiated (+) spermatocytes. The sum of the area stained for the Dmc1/Rad51 foci in each nucleus was measured in wild-type nuclei at leptonema (L), early zygonema (EZ), late zygonema (LZ), and pachynema (P), in leptotene- or early zygotene-like (L/EZ-like) *spo11*[-/-], and in early zygotene-like (EZ-like) *sycp2*[-/-] spermatocytes. Numbers at the bottom (n) indicate numbers of nucleus measured in each data set. Center lines show the medians; box limits indicate the 25th and 75th percentiles as determined by R software; whiskers extend 1.5 times the interquartile range from the 25th and 75th percentiles; data points are plotted as open circles. Chromosomal spreads of four (wild-type) and two (*spo11*[-/-] and *sycp2*[-/-]) individual fish were used for each condition. *** indicates p<0.0001 (Student's t-test). N.S. indicates not significant.
(TIF)

**S12 Fig. A model of early meiotic events in wild-type and *sycp2* knockout zebrafish spermatocytes.** In zebrafish spermatocytes, Sycp3 is expressed in aggregates before leptonema (i; preleptonema), and telomeres are observed in bouquets. In leptonema, upon axis formation of Sycp2 near telomeres, Sycp3 is also localized on an axis (ii). In the same stage, DSB formation also occurs in the proximity of the telomeres, as indicated by Rad51 [26], Dmc1/Rad51 and RPA staining. In zygonema, synapsis is also initiated near telomeres by the localization of Sycp1 (iii). Synapsis is completed at pachynema, and telomeres are dissociated from bouquets at this time (iv) [26]. When we knocked out Sycp2, the signals of DSB markers were rarely detected, and homologous pairing was strongly impaired (v).
(TIF)

**S13 Fig. Conservation of domain structures between mouse SYCP2, zebrafish Sycp2 and budding yeast Red1.** A: Schematic presentations of domain structures of mouse SYCP2 (mSYCP2), zebrafish Sycp2 (zSycp2) and budding yeast Red1. Domain structures of mSYCP2 and Red1 were from [19]. NTD, N-terminal domain; HORMA/Hop1-BD, putative HORMA/Hop1-binding domain; CC, C-terminal coiled-coil domain. The numbers in the panel correspond to the positions of the amino acid residue. C-terminal regions of mSYCP2 that are involved in interactions with SYCP1 (ID-SYCP1) [23] and SYCP3 (ID-SYCP3) [22] are shown in green and yellow bars, respectively. A C-terminal region truncated in the previously published Sycp2 mutant mouse (SYCP2t) is also shown [22]. Percent identity/similarity between each domain of zSycp2 and that of mSYCP2/Red1 is shown at the bottom. B: The alignment of HORMA/Hop1-binding domains (HORMA/Hop1-BD) across species. Alignment was performed with sequences of putative HORMA/Hop1-BDs from *Homo sapiens* (Hs), *Mus musculus* (Mm), *Alligator mississippiensis* (Am), *Danio rerio* (Dr), *Callorhinchus milii* (Cm), *Zygosaccharomyces rouxii* (Zr), and *Saccharomyces cerevisiae* (Sc) by the T-coffee program. Conserved residues with similar properties were visualized by the MView program. Reference IDs of each sequence are indicated in the panel.
(TIF)

**S14 Fig. Ovary and testis morphologies used to evaluate sexual phenotypes of 8- to 9-week-old zebrafish.** A: A light field image of zebrafish testis. The region outlined with black is shown at a higher magnification at the bottom. Spermatids and/or sperm cells are outlined with broken lines. B: A light field image of zebrafish ovary with an oocyte outlined in broken lines.
(TIF)

**S1 Table. List of genes in the causal genomic locus of *its* mutant zebrafish.**
(TIF)

**S2 Table. Sequencing of the exon 8–9 splice site of *sycp2* cDNA.**
(TIF)

**S3 Table. Primers used in this study.**
(TIF)

**S4 Table. Antibodies used in this study.**
(TIF)

**S5 Table. Numerical data.**
(XLSX)

## Acknowledgments

We thank Kazuhiro Maeshima and Satoru Ide for advice on telomere polyamide staining; James Amatruda for the anti-γH2AX antibody; Kellee R. Siegfreid for the *its* mutant zebrafish line; and Minori Shinya for the TALEN vectors. The generation of the anti-Sycp2 and anti-Rad21l1 antibodies was supported by a program of the Joint Usage/Research Center for Developmental Medicine, Institute of Molecular Embryology and Genetics, Kumamoto University. The γ-ray irradiation experiments were conducted through the Joint Usage/Research Center Program of the Radiation Biology Center, Kyoto University. We thank Yasushi Hiromi and Mitsuhiko Kurusu, URA members at the National Institute of Genetics, for critical reading and discussions.

## Author Contributions

**Conceptualization:** Kazumasa Takemoto, Yukiko Imai, Kenji Saito, Noriyoshi Sakai.

**Data curation:** Kazumasa Takemoto, Yukiko Imai, Noriyoshi Sakai.

**Formal analysis:** Kazumasa Takemoto, Yukiko Imai.

**Funding acquisition:** Yukiko Imai, Kei-ichiro Ishiguro, Noriyoshi Sakai.

**Investigation:** Kazumasa Takemoto, Yukiko Imai, Kenji Saito.

**Methodology:** Kazumasa Takemoto, Yukiko Imai, Kenji Saito, Toshihiro Kawasaki, Peter M. Carlton, Kei-ichiro Ishiguro, Noriyoshi Sakai.

**Project administration:** Noriyoshi Sakai.

**Resources:** Peter M. Carlton, Kei-ichiro Ishiguro, Noriyoshi Sakai.

**Software:** Yukiko Imai.

**Supervision:** Yukiko Imai, Peter M. Carlton, Kei-ichiro Ishiguro, Noriyoshi Sakai.

**Validation:** Kazumasa Takemoto, Yukiko Imai, Kenji Saito.

**Visualization:** Yukiko Imai, Noriyoshi Sakai.

**Writing – original draft:** Yukiko Imai, Noriyoshi Sakai.

**Writing – review & editing:** Kazumasa Takemoto, Yukiko Imai, Kei-ichiro Ishiguro, Noriyoshi Sakai.

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
