## [Decision Letter · Decision Letter 0]

11 Aug 2019

Dear Dr Sakai,

Thank you very much for submitting your Research Article entitled 'Sycp2 is essential for synaptonemal complex assembly and early meiotic recombination in zebrafish spermatocytes' to PLOS Genetics. Your manuscript was fully evaluated at the editorial level and by independent peer reviewers. The reviewers appreciated the attention to an important problem, but raised some substantial concerns about the current manuscript. Based on the reviews, we will not be able to accept this version of the manuscript, but we would be willing to review again a much-revised version. We cannot, of course, promise publication at that time.

Should you decide to revise the manuscript for further consideration here, your revisions should address the specific points made by each reviewer. We encourage you to pay particular attention to the excellent suggestions from Reviewers 1 and 3, and also we urge you to take into account, and to discuss in detail your data in the context of the recently published work from Sean Burgess and colleages in this journal. We will also require a detailed list of your responses to the review comments and a description of the changes you have made in the manuscript.

If you decide to revise the manuscript for further consideration at PLOS Genetics, please aim to resubmit within the next 60 days, unless it will take extra time to address the concerns of the reviewers, in which case we would appreciate an expected resubmission date by email to plosgenetics@plos.org.

[LINK]

We are sorry that we cannot be more positive about your manuscript at this stage. Please do not hesitate to contact us if you have any concerns or questions.

Yours sincerely,

Paula E. Cohen

Associate Editor

PLOS Genetics

Gregory P. Copenhaver

Editor-in-Chief

PLOS Genetics

Reviewer's Responses to Questions

**Comments to the Authors:**

Reviewer #1: The the zebrafish Ietsugu (its) mutant was previously identified by the authors in a forward genetic screen for male sterility mutants and was found to be defective in generating the meiotic chromosome axis. In this study, the researchers map the its mutation to the sycp2 locus and show it is a T to A substitution that causes aberrant splice site usage. The its mutation behaves as a hypomorphic allele with expression levels ~26% of wild type. The authors created a TALEN generated knockout of the sycp2 gene and find it is required for axis formation (i.e. Sycp3 staining) and DSB formation (Dmc1 staining), but that some SC (Sycp1) forms ectopically. The zebrafish sycp2-/- mutant phenotypes are comparable to a SYCP2-/- mouse. Several new resources were developed including antibodies raised to zfDmc1 and zfSycp2 that could be useful for the burgeoning zebrafish meiosis field.

The work describing the sycp2 phenotypes is solid and the immunofluorescence figures are very nice and support the novel conclusions of the paper. In addition, having a hypomorphic allele in hand adds to the study. As zebrafish is an emerging model for meiosis, understanding the evolutionary conservation of function of potential orthologs is important so identifying the its mutation as an allele of Scyp2 is significant.

There are several issues that need to be addressed before publication. Several conclusions made by the authors are either not supported by the data or are not based on novel observations (e.g. Sycp2 as a potential functional ortholog of yeast Red1, DSBs occur near telomeres in zebrafish, zebrafish as a model for human male meiosis). The comments below suggest ways that the paper can be better focused to highlight the novel aspects of the work.

Comments:

1. The authors suggest that they have discovered Sycp2 as a functional ortholog of yeast Red1 protein since pairing, DSB formation, axis formation and SC assembly are defective in the sycp2-/- mutant. Reporting these data and discussing its relevance to Red1 is important, but arguing that they have made the novel discovery that Sycp2 is the functional Red1 ortholog is a bit of an overstatement. The link between Red1 and Sycp2 has been explored previously in mouse studies and recent structural analysis has revealed commonalities and differences between the two proteins (West, eLIFE 2019; Feng, 2107). Red1 protein forms a homotetramer while mouse SYCP2/SYCP3 forms an analogous heterotetrameric structure. One of the main arguments the authors make supporting the notion that Scyp2 is a potential ortholog of Red1 stems from the failure to see DSBs in the zebrafish sycp2 mutant and previous reports that DSBs form in the mouse SYCP3-/- mutant. This is rather circumstantial evidence. To really nail this down, the authors would need to create the zebrafish sycp3-/- mutant which seems beyond the scope of the current paper. The authors should also be aware that yeast red1 mutant does in fact form DSBs, but they are repaired with the sister chromatid (Blat et al, 2002). Most of the discussion on pages 16 and 17 could be cut so that the authors say briefly that their findings are consistent with the notion that Scyp2 is the functional ortholog of Red1.

2. The authors suggest that Sycp2 is required for ovarian development in zebrafish based on a sex reversal phenotype described in a previous publication (pages 15-16). Granted, other mutations affecting oogenesis have been shown to cause sex reversal, but these studies have also shown that sex reversal is suppressed by tp53-/- mutation, further suggesting a defect in meiotic chromosome dynamics during oogenesis. To make the claim that sex reversal is due to a check point arrest, the scyp2-/- tp53-/- double mutant should be made.

3. In addition, there should be some documentation on how many fish were analyzed for a sex reversal phenotype and the criteria used for sexing fish. Without directly scoring gonads with arrested oocytes, it is a bit of an overstatement to conclude that Scyp2 is required for ovarian development. It would be important to know if some small fraction of females escape sex reversal. Perhaps females arise more frequently in the its mutant? If so then analysis of ovarian development could be carried out on those females. If the tp53-/- mutation suppresses the sex reversal phenotype then there should be a population of females where ovarian development could be analyzed. Exploring this phenotype in more detail would be interesting since the phenotype is markedly different from the spo11 mutant, which also fails to pair or synapsis chromosomes, but produces fertile oocytes.

4. The authors also elaborate on their observations that DSBs form near telomeres in males (pages 14-15). This is not a novel observation and has been reported in other zebrafish studies, including a paper by the author (Saito et al, Dev Dyn 2011, 2014; Sansam and Pezza 2015 (not cited) and Blokhina et al, 2019). This section should be shortened to focus the discussion on the role of Sycp2 in DSB formation.

5. The formation of SC stretches in the absence an axis is interesting. Perhaps the authors could speculate how the SC is forming without an axis. Is there a precedence for this? Could they be forming between sister chromatids?

Minor points:

Line 173: “Sycp1 began to be observed as short fragments followed by the appearance of Sycp3 and Sycp2 signals”. Is this correct? The figure looks like Sycp1 forms fragments after Scyp3 and Scyp2.

Reviewer #2: The Manuscript of Takemoto et al presents the characterization of SYCP2 in zebrafish through the cloning of the its locus as a splice-site mutation in the sycp2 gene. The creation of additional alleles with similar phenotypes and the development of spreading techniques allow the authors to demonstrate defects in pairing, synapsis, and DSB formation/early repair intermediate processing and the interdependencies with SYCP3 and SYCP1.

The experiments are clearly presented and well controlled and the elucidation of SYCP2 functions is novel. A number of the details of wild type meiosis and the characterization of pairing at telomeres are partially redundant with the recent work from Sean Burgess’ lab, although some of the approaches are complementary rather than redundant.

Overall the paper could benefit from a bit of reorganization. Since SYCP2 is an axis proteins, it seems that it would make sense to present the role of mutation in pairing and then synapsis and DSB formation.

Given the Burgess paper has recently described staging of the events of meiosis with SYCP3 staining and telomere probes, it would behoove the authors to spend less time on the naming of the stages rather than the differences seen in the sycp2 mutants.

In the discussion of sex reversal, the authors should consider the possibility that the SYCP2 and may activate checkpoints exclusively in the female that eliminate oocytes (and female development), allowing for male development. Differences in meiotic checkpoint activation is a feature of numerous species.

Related to this, the difference between spo11 and sycp2 mutants vis-à-vis sex reversal suggests either 1) that DSBs are made in sycp2 but are incapable of recruiting DMC1 and H2AX (this could potentially be testing by irradiating the fish, as is often down in worm meiosis to show DSB rescue) or 2) that a synapsis checkpoint is killing off female cells. Given the uniqueness of these phenotypes, it would be valuable to add these details to the main text and perhaps examine checkpoints in the fish, if possible. Indeed no mention is made of a female phenotype even in the weak loss of functions. Is there one?

Minor comments:

In leptonema in Fig. 3A, it is unclear what the upper right corner is. Is this a different cell? A zoom in of a part of the image?

Line 173, it is unclear what is meant by “followed by” since they are fixed images and the stains are superimposable.

For continuity, in Figure 4, SYCP3 should be on top and SYCP1 below (similar to figure 3).

It would be helpful to label Figure 4A as wild type.

Figure S6 B should indicate in labels colocalization with telomere probes.

Reviewer #3: The synaptonemal complex is a meiosis-specific chromosome structure that is critical for the execution of recombination during gamete development. While previous studies in mammals have focused on the effects of chromosome axis component Sycp2 on synaptonemal complex assembly and synapsis, its role in DSB formation has not yet been investigated in mammals. As evidenced by recent papers (including Blokhina et al PLoS Genetics 2019), the use of zebrafish as a model system for studying meiosis has proven powerful for understanding meiotic processes in organisms that initiate SC formation in subtelomeric regions (e.g. humans). In the following manuscript by Takemoto et al., the authors have identified an allele of the conserved synaptonemal complex protein Sycp2 through a previously published forward genetic screen for zebrafish sterility phenotypes (Saito et al., Developmental Dynamics 2011). The authors convincingly demonstrate that this allele acts as a hypomorph of variable penetrance arising from a splicing defect due to a mutation within exon 8 of sycp2. Further, the authors utilized TALEN mutagenesis to generate a null allele of sycp2 and characterized its meiotic phenotypes. Using the combination of these two alleles in zebrafish, the authors clearly show that Sycp2 is required for sperm production in males. The lack of sperm production appears to be due to defects that occur during meiotic prophase I: assembly of the synaptonemal complex, pairing of homologs, and meiotic DSB formation. Based on these phenotypes and the fact Sycp2 is a meiotic chromosome structure component, the authors provide support for the conclusion from West et al., eLife 2019 that Sycp2 serves as an axis component analogous to the well-studied Red1 in S. cerevisiae and HTP-3 in C. elegans. Further, similar to spo11 and other meiotic null mutations, the authors find that loss of sycp2 prevents the development of viable female zebrafish. Overall, the evidence that the authors provide for the role of Sycp2 in assembly of the SC and DSB formation is convincing. Taken together, the authors provide a good characterization of two alleles of a known meiotic chromosome axis protein homolog.

Comments to improve the conclusions and impact of the manuscript:

1. The authors qualitatively show a decrease in meiotic DNA damage by immunofluorescence using Dmc1 and gamma-H2AX; however, the amount of DNA damage they observe in sycp2 mutants appears to be lower than that of wild-type, but not entirely eliminated (e.g. images in Figure 6A-iv and 6B-iv for Dmc1). The extent of DNA damage (via number of Dmc1 foci) in wild-type, sycp2, and spo11 mutants should be quantified to indicate the exact severity of the mutant phenotypes relative to each other and wild-type.

2. Using quantification of telomere foci in fixed cells, the authors suggest that chromosome pairing at the telomeres is partially impaired in the sycp2 hypomorph. As mutants for various meiotic chromosome structures have been shown in several systems to lead to nonhomologous pairing, the authors should definitively demonstrate whether homologous pairing is occurring at all in this mutant by using fluorescence in situ hybridization targeting an autosomal locus, as performed in a recent PLoS Genetics publication (Blokhina et al 2019). This result would provide important, strong validation of their conclusion and eliminate the possibility that nonhomologous pairing is occurring in this mutant, as occurs in chromosome structure mutants in other systems.

3. In the present study, the authors find an effect on meiotic recombination initiation by looking at Dmc1 staining in sycp2 mutants. Previous studies by Doug Bishop’s lab and Akira Shinohara’s lab have established that in S. cerevisiae, Rad51 is required for Dmc1 foci formation. To both further validate the finding of this manuscript that recombination initiation is inhibited and add additional mechanism to the action of Sycp2 in the early stages of meiotic recombination, the authors to can evaluate Rad51 foci localization (using the Rad51 antibody from Blokhina et al., PLoS Genetics 2019) together with Dmc1 localization in wild type, spo11, and sycp2 mutants. Further, this co-localization experiment would be useful to determine whether Dmc1 and Rad51 colocalize (and the frequency of their colocalization) in zebrafish.

4. In the discussion, the authors draw comparisons between Sycp2 homologs in mice, yeast, nematodes, humans, and zebrafish. Supplementing this discussion with a specific bioinformatic analysis of conserved domains and sequence of zebrafish Sycp2 and Red1 (as was performed for the C-terminus of Red1 and mouse Sycp2 in West et al., eLife 2019) would strengthen their conclusion as well as possibly provide interesting insights into the components of synaptonemal axis proteins required for performing diverse roles in meiotic prophase I.

5. For Figure 3B, please indicate in both the image and figure legend what part of the image the inset panel is for the leptotene images.

6. For Figure 5B, several changes would increase the clarity of this figure. First, please include the genotypes and stages of the images in the Figure itself. Second, the labels for the three- and four-color images are confusing, especially the label for Sycp3 (perhaps put the labels, with Sycp3 in white, all against a black background to clearly indicate that Sycp3 is in white in the images). Third, include better zoom-in panels of these images similar to Blokhina et al., PLoS Genetics 2019, they are zoomed in at a higher level with clear colors that enable the reader to see individual DSB sites along the axis near the telomeres.

7. The ladder for the RT-qPCR in Figure S4B appears to be from a different gel than the result of the lanes in the gel; however, there is no indication either the presentation of the image or the figure legend that this is the case. This panel as well as the figure legend should be amended appropriately.

8. The western blot in Figure S4C appears very overexposed as well as extremely adjusted with post-processing methods outside of the dynamic range. Inclusion of panels from under/over exposed images of the blot as well as without any image adjustment should be included for this figure.

9. To improve clarity of the figure, Figure S5 should have the different meiotic stages indicated for the images.

10. In Figure S7, the DAPI levels for the spo11-/- mutant appears to be decreased significantly relative to the wild type images. Please check the exposures of the channel for these images.

11. Several of the findings of the authors have already been found in an earlier published study (Blokhina et al., PLoS Genetics 2019). For example, the authors include in their abstract that they demonstrate SC assembly occurs in the vicinity of telomeric regions (p. 1 line 12) and that DSB formation occurs adjacent to chromosome ends (p. 4 lines 87-88). As these results has already been shown by Blokhina et al., PLoS Genetics 2019, I suggest the authors remove these statements in the abstract and last paragraph of the intro where they present them as novel findings, and instead add stronger definition/emphasis of their own novel results from those results that were also found in the recent Burgess lab study.

**Have all data underlying the figures and results presented in the manuscript been provided?**

Reviewer #1: No: There is no documentation of the sex-reversal phenotype

Reviewer #2: Yes

Reviewer #3: Yes

PLOS authors have the option to publish the peer review history of their article (what does this mean?). If published, this will include your full peer review and any attached files.

Reviewer #1: No

Reviewer #2: No

Reviewer #3: No

---

## [Decision Letter · Decision Letter 1]

16 Dec 2019

Dear Dr Sakai,

Thank you very much for submitting your Research Article entitled 'Sycp2 is essential for synaptonemal complex assembly, early meiotic recombination and homologous pairing in zebrafish spermatocytes' to PLOS Genetics. Your manuscript was fully evaluated at the editorial level and by independent peer reviewers. The reviewers appreciated the attention to an important topic but identified some aspects of the manuscript that should be improved. In particular, Reviewers 2 and 3 were particularly adamant that clearly defined and simple experiments have been omitted from this work, and this substantially weakens the impact of what otherwise could be a strong story.

We therefore ask you to modify the manuscript according to the review recommendations before we can consider your manuscript for acceptance. Your revisions should address the specific points made by each reviewer.

[LINK]

Yours sincerely,

Paula E. Cohen

Associate Editor

PLOS Genetics

Gregory P. Copenhaver

Editor-in-Chief

PLOS Genetics

Reviewer's Responses to Questions

**Comments to the Authors:**

Reviewer #1: The authors did a wonderful job addressing the comments and making the story richer with more data. Overall is a very nice paper.

Minor comments:

The first section of the discussion on the hypomorphic allele of sycp2its is redundant with the results. I think it can be left out completely since much of the discussion is already in the results section. It takes away from the important meiotic phenotypes the authors want to highlight.

Seeing 3 and 4 BAC probe spots in the sycp2 mutant indicates there may be a sister chromatid cohesion defect as the authors say. The foci are really far away from each other. Do you think the sister chromatids are completely separated from each other since there is not axis for them to anchor? It will take more experiments with multiple BAC probes to figure this out, but I was surprised. What does an sycp3 mouse mutant look like? Is that informative?

Reviewer #2: Re-review: Sycp2 is essential for synaptonemal complex assembly, early meiotic recombination

and homologous pairing in zebrafish spermatocytes

Overall manuscript is much tighter and easier to read and the authors have addressed a number of the issues raised by the reviewers.

1) The main issue they do NOT address is the role of the checkpoint in the loss of female embryos. In multiple places they said “in the interest of time” However, the paper was submitted initially in July and returned. If they had done the experiment it would have delayed another 3 months, true, but this is one of the novel aspects of the work and would indeed answer a critical question.

In the description of this arrest, the text is unclear for non-zebrafish readers. On p.8, in the observation that there are no female offspring and that it is required for ovarian development, the link between ovarian development and sex determination needs to be better described. One suggested is to add the text that was provided as a response to the reviewers:

“We speculate that the checkpoint is rather active in both oogenesis and

spermatogenesis in sycp2-/- zebrafish. In zebrafish, all juveniles develop gonads with

immature oocytes regardless of their definitive sex. The individual in which these

immature oocytes degenerate becomes male. Therefore, zebrafish mutants depleted of

oocytes develop as males that are mostly infertile. In sycp2-/- juvenile gonads, we

observed previtellogenic dictyate oocytes in four among five individual fish (S10 Fig).

Therefore, the absence of females in sycp2 mutant fish (Table 1) implies that oocytes

were eliminated at the juvenile age, possibly due to checkpoint activation in oogenesis.

As a result, all sycp2 mutant fish develop as males.”

2) In response to comment #4 about distinguishing whether the lack of DMC1 foci is a defect in DSBs or early DSB processing, the authors claim they did not have time to find the right dose of IR. This is actually a poor excuse: if the endpoint is sterility, this is an issue, but they could easily have looked for RAD51 and DMC1 foci within 30 minutes of irradiation to check whether these were detectable. This is still a do-able and important experiment.

Text:

Rpa foci should be RPA foci.

Line 362: I would say that in the fanc1 and brca2 mutants that they became “sterile males”

Reviewer #3: The authors have made a good effort to address reviewer concerns. Their work is strengthened by their removal and deemphasis of their identical results to that already published in Blokhina et al PLoS Genetics 2019. Further, their addition of data in Figure 5 to demonstrate pairing defects in the sycp2 mutant alleles also strengthens the paper. It is unfortunate that many of the comments of all three reviewers were unable to be addressed due to lack of time. The addition of several of the requested experiments, such as assessing rescue of DSBs and fertility by irradiation of sycp2 mutants. With regards to some of their responses to reviewers, I have two major comments/concerns that decrease the impact of their work and strength of their conclusions:

1) The authors mention in their response to Reviewer#3 comment #3 that their Dmc1 “antiserum was raised against a part of the Dmc1 protein that is 43% identical to the zebrafish Rad51. Thus, our anti-Dmc1 antiserum also potentially recognized Rad51.” This fact regarding their Dmc1 antibody is troubling because it may not be specific to Dmc1. The authors should either 1) amend their discussion of these results and their figures to reflect the fact their Dmc1 antibody may actually be a Dmc1/Rad51 antibody (all reference to Dmc1 should be changed to Dmc1/Rad51); or, 2) they should directly determine whether their antibody is specific to Dmc1 by looking at localization in a dmc1 null mutant by IF and/or Western.

2) In response to Reviewer #1 comment #5, the authors look at Rad21l1l localization with Sycp1 stretch to determine whether the SC is forming between sisters. Although they see colocalization of Rad21l1l and Scyp1, this is not a definitive determinant of whether Sycp1 is forming between sisters because Sycp1 could be forming along a single sister chromatid axis. In order to determine if Sycp1 is forming between sisters, the two sister chromatid axes need to be resolved and Sycp1 needs to be see forming between those two axes. The use of DeltaVision deconvolution microscopy is not at a sufficient resolution level to resolve two axes. The authors need to use SIM microscopy or EM to determine the formation of SC between two sister chromatid axes (as was done in Cahoon CK, Helm JM, and Libuda DE Genetics 2019 and Xu H, Beasley MD, … McKay MJ Dev Cell 2005 respectively). Alternatively, the authors could remove this data and simply postulate in the discussion that Sycp1 forming between sister chromatids in this context.

**Have all data underlying the figures and results presented in the manuscript been provided?**

Reviewer #1: Yes

Reviewer #2: Yes

Reviewer #3: Yes

PLOS authors have the option to publish the peer review history of their article (what does this mean?). If published, this will include your full peer review and any attached files.

Reviewer #1: No

Reviewer #2: No

Reviewer #3: No

---

## [Editor Report · Decision Letter 2]

29 Jan 2020

Dear Dr Sakai,

We are pleased to inform you that your manuscript entitled "Sycp2 is essential for synaptonemal complex assembly, early meiotic recombination and homologous pairing in zebrafish spermatocytes" has been editorially accepted for publication in PLOS Genetics. Congratulations!

Yours sincerely,

Paula E. Cohen

Associate Editor

PLOS Genetics

Gregory P. Copenhaver

Editor-in-Chief

PLOS Genetics

Comments from the reviewers (if applicable):

**Data Deposition**

http://datadryad.org/submit?journalID=pgenetics&manu=PGENETICS-D-19-00970R2

**Press Queries**

---

## [Editor Report · Acceptance letter]

20 Feb 2020

PGENETICS-D-19-00970R2 

Sycp2 is essential for synaptonemal complex assembly, early meiotic recombination and homologous pairing in zebrafish spermatocytes 

Dear Dr Sakai, 

We are pleased to inform you that your manuscript entitled "Sycp2 is essential for synaptonemal complex assembly, early meiotic recombination and homologous pairing in zebrafish spermatocytes" has been formally accepted for publication in PLOS Genetics! Your manuscript is now with our production department and you will be notified of the publication date in due course.

With kind regards,

Matt Lyles

PLOS Genetics

On behalf of:
